# THREE MECHANISMS OF FEATURE LEARNING IN A LINEAR NETWORK

**Yizhou Xu[1], Liu Ziyin[2,3]**

[1]*Computer and Communication Sciences, École Polytechnique Fédérale de Lausanne*
[2]*Research Laboratory of Electronics, Massachusetts Institute of Technology*
[3]*Physics & Informatics Laboratories, NTT Research*

## ABSTRACT

Understanding the dynamics of neural networks in different width regimes is crucial for improving their training and performance. We present an exact solution for the learning dynamics of a one-hidden-layer linear network, with one-dimensional data, across any finite width, uniquely exhibiting both kernel and feature learning phases. This study marks a technical advancement by enabling the analysis of the training trajectory from any initialization and a detailed phase diagram under varying common hyperparameters such as width, layer-wise learning rates, and scales of output and initialization. We identify three novel prototype mechanisms specific to the feature learning regime: (1) learning by alignment, (2) learning by disalignment, and (3) learning by rescaling, which contrast starkly with the dynamics observed in the kernel regime. Our theoretical findings are substantiated with empirical evidence showing that these mechanisms also manifest in deep nonlinear networks handling real-world tasks, enhancing our understanding of neural network training dynamics and guiding the design of more effective learning strategies.

## 1 INTRODUCTION

It has been shown that for a neural network under certain types of scaling towards infinite width (or certain parameters), the learning dynamics can be precisely described by the neural tangent kernel (NTK) dynamics (Jacot et al., 2018), or the "kernel regime." We say that a model is in the kernel regime if the NTK of the model remains unchanged throughout training, and the learning dynamics is linear in the model parameters. When the learning dynamics is not linear, we say that the model is in the feature learning regime. Since then, a lot of works have been devoted to the study of how the kernel evolves during training as it sheds light on nonlinear mechanisms of learning (Liu et al., 2020; Huang et al., 2020; Huang & Yau, 2020; Chen et al., 2020; Baratin et al., 2021; Atanasov et al., 2021; Geiger et al., 2021; Bordelon & Pehlevan, 2022; Simon et al., 2023).

Despite this progress, a comprehensive theory that accurately characterizes both the kernel and feature learning dynamics in finite-width models remains elusive. Most existing works focus on infinite-width settings, where the behavior of the network simplifies, but do not extend well to finite configurations. This gap leaves several theoretical and practical questions about the fundamental nature of learning in neural networks unanswered. Specifically, there is a lack of analytically solvable models that can exhibit both NTK and feature learning dynamics, which are crucial for understanding how real-world neural networks learn and adapt. Our main contributions are

- We analytically solve the evolution dynamics of the NTK for a minimal finite-width model with arbitrary initialization. The model we analyze is a one-hidden-layer linear network, which, despite its simplicity, has a non-convex loss landscape and strongly coupled dynamics. Prior to our work, the exact solution for its learning dynamics was unknown.
- Based on our exact solutions, we identify three novel mechanisms of learning that are exclusive to the feature learning phase of the network.
- Using our exact solutions, we provide phase diagrams that distinguish between the kernel phase and the feature learning phase, for both finite and infinite width models.

---

[*]Authors are listed alphabetically.

- We empirically validate the three mechanisms of feature learning and our phase diagrams in realistic nonlinear networks.

This work is structured as follows. We introduce related literature in Section 2 and Appendix A. Our model and solution are presented in Section 3.1, followed by three mechanisms of feature learning in Section 3.2 and 3.3. Section 4 gives the phase diagram that separates feature learning phase and kernel phase. Proofs are provided in Appendix B. Experimental details are given in Appendix C.

## 2 RELATED WORK

**Kernel and feature learning**. Under the NTK scaling, it is shown that NTK remains unchanged in the infinite-width limit (Jacot et al., 2018; Lee et al., 2019; Arora et al., 2019b), where the network is asymptotically equivalent to the kernel regression using NTK. Higher order feature learning corrections of the NTK have also been studied (Hanin & Nica, 2019; Dyer & Gur-Ari, 2019; Andreassen & Dyer, 2020; Roberts et al., 2022). An alternative to the NTK parameterization is the mean-field (or $\mu P$) parameterization where features evolve at infinite width (Mei et al., 2018; Yang & Hu, 2020; Bordelon & Pehlevan, 2022). Within this literature, the works closest to ours are those computing finite width corrections (Pellegrini & Biroli, 2020; Pham & Nguyen, 2021; Bordelon & Pehlevan, 2024). However, these results are perturbative in nature and applicable when the width is large. Our study has the same goal of understanding the learning dynamics but with a different approach. We analytically solve a model that admits analysis both when the model size is finite and infinite.

**Linear networks**. Linear networks have been extentively used as toy models to understand complex dynamics of nonlinear networks. For example, they provide significant insights into the loss landscape (Baldi & Hornik, 1989; Ziyin et al., 2022), optimization (Saxe et al., 2013; Huh, 2020; Tarmoun et al., 2021; Braun et al., 2022), generalization (Lampinen & Ganguli, 2018; Gunasekar et al., 2018) and learning dynamics (Arora et al., 2018; 2019a; Ziyin et al., 2024) of neural networks. Closely related to ours are Saxe et al. (2013), Braun et al. (2022), and Atanasov et al. (2021), which solve the learning dynamics of linear models under special initializations. Mathematically, previously known results are *particular* solutions to the differential equation, whereas our solution is a *general* solution. A contemporary paper (Kunin et al., 2024) studies the exact solution in case of one hidden neuron, whereas our result applies to arbitrary number of neurons. Another contemporary paper (Beneventano & Woodworth, 2025) analyzes the finite stepsize effect for the same model.

## 3 AN EXACTLY SOLVABLE MODEL

### 3.1 PROBLEM SETTING AND SOLUTION

Let us consider a two-layer network with power activations $f(\boldsymbol{x}) = \gamma \sum_{i=1}^{d} u_i (\sum_{j=1}^{d_0} w_{ij} x_j)^{\beta}$, where $d_0$ is the input size, $d$ is the network width, $\boldsymbol{u}$ and $W$ are the weight vector/matrix of the second and the first layer, respectively, and $\gamma$ is a normalization factor. We first show that the learning dynamics of this network can be reduced to a 1d dynamics for any $\beta$, and then solve the dynamics exactly for the linear network case where $\beta = 1$.

We consider the following network trained on the MSE loss:

$$\tilde{L}(\mathbf{u}, W) = \mathbb{E}_{\tilde{x}}\left[ (\gamma \sum_{i=1}^{d} u_i (\sum_{j=1}^{d_0} w_{ij} \tilde{x}_j)^{\beta} - y(\tilde{x}))^2 \right], \tag{1}$$

where we treated the target $y$ as a function of $\tilde{x}$. $\mathbb{E}$ denotes the averaging over the training set. The training proceeds with the gradient flow algorithm. We allow the two layers to have different learning rates, $\eta_u$ and $\eta_w$:

$$\frac{du_i}{dt} = -\eta_u \frac{\partial \tilde{L}}{\partial u_i}, \quad \frac{dw_{ij}}{dt} = -\eta_w \frac{\partial \tilde{L}}{\partial w_{ij}}. \tag{2}$$

We restrict to when the data lies on a 1d subspace, and the following proposition shows that the learning dynamics under Eq.(1) is equivalent to that under a simplified loss.

**Proposition 1.** *Let $\tilde{x} = an$, where $a \in \mathbb{R}$ is a random variable and $n$ is a fixed unit vector. Let $x = \sqrt{\mathbb{E}[\tilde{x}^2]} n$ and $y = \frac{\mathbb{E}[\tilde{x}y(\tilde{x})]}{\sqrt{\mathbb{E}[\tilde{x}^2]}}$. Then, the gradient flow of Eq.(1) equals the gradient flow of*

$$L(\mathbf{u}, W) = \left[ \gamma \sum_{i=1}^{d} u_i (\sum_{j=1}^{d_0} w_{ij} x_j)^{\beta} - y \right]^2, \tag{3}$$

The following theorem gives a precise characterization of the dynamics of $\mathbf{u}$ and $W$ for arbitrary initialization and hyperparameter choices.

**Theorem 1.** *Let*

$$
\begin{cases}
p_i(t) := \frac{1}{2\rho}\left(\sqrt{\eta_u}\sum_{j=1}^{d_0} w_{ij}(t)x_j + \sqrt{\beta\eta_w}\rho u_i(t)\right), \\
q_i(t) := \frac{1}{2\rho}\left(\sqrt{\eta_u}\sum_{j=1}^{d_0} w_{ij}(t)x_j - \sqrt{\beta\eta_w}\rho u_i(t)\right),
\end{cases}
\tag{4}
$$

*where $\rho := \sqrt{\frac{1}{d_0}\sum_{i=0}^{d_0} x_i^2}$.*

*1. (**Analytic Reduction to 1d Dynamics**) The solution of the gradient flow can be given by the solution of the following ODE*

$$
\frac{d\Delta_i(t)}{dt} = -2\gamma\sqrt{\beta\eta_u^{2-\beta}\eta_w}\rho^\beta\left(\gamma\rho^\beta(\beta\eta_u^\beta\eta_w)^{-1/2}\sum_{j=1}^d (p_j(t) - c_j/p_j(t))(p_j(t) + c_j/p_j(t))^\beta - y\right),
\tag{5}
$$

*with $p_j(t) = F_j^{-1}(\Delta_i(t) - \Delta_i(0) + \Delta_j(0))$ for all $i, j = 1, 2, \cdots, d$, where*

$$
F_i(x) := \int \frac{dx}{x(x + c_i/x)^{\beta-1}},
\tag{6}
$$

*$F_i^{-1}$ denotes the inverse function of $F_i$, $c_i := p_i(0)q_i(0)$. The initial conditions are $\Delta_i(0) = F_i(p_i(t))$.*

*After solving $\Delta_i(t)$, the weight vector/matrix are given by*

$$
\begin{cases}
u_i(t) = \frac{1}{\sqrt{\beta\eta_w}}(p_i(t) - q_i(t)), \\
w_{ij}(t) = w_{ij}(0) + (p_i(t) + q_i(t))\frac{x_j}{\sqrt{\eta_u}\rho},
\end{cases}
\tag{7}
$$

*where $p_i(t) = F_i^{-1}(\Delta_i(t))$, $q_i(t) = \frac{c_i}{p_i(t)}$.*

*2. (**Exact Solution of Linear Net**) Let $P := \frac{1}{d}\sum_{j=1}^d p_j(0)^2$, $Q := \frac{1}{d}\sum_{j=1}^d q_j(0)^2$. For $\beta = 1$, if $P \neq 0$, the solution is*

$$
\begin{cases}
p_i(t) = p_i(0)\left[\frac{\alpha_+ + \xi(t)\alpha_-}{1 - \xi(t)}\right]^{1/2}, \\
q_i(t) = q_i(0)\left[\frac{\alpha_+ + \xi(t)\alpha_-}{1 - \xi(t)}\right]^{-1/2},
\end{cases}
\tag{8}
$$

*where[1]*

$$
\xi(t) := \frac{1 - \alpha_+}{1 + \alpha_-}\exp\left(-4t/t_c\right),
\tag{9}
$$

$$
t_c := 1/\left(\sqrt{\eta_u\eta_w\gamma^2\rho^2 y^2 + 4\rho^4(\gamma^2 d)^2 PQ}\right),
\tag{10}
$$

$$
\alpha_\pm := \frac{1}{2(\gamma^2 d)\rho^2 P}\left(\sqrt{\eta_u\eta_w}\gamma\rho y \pm t_c^{-1}\right).
\tag{11}
$$

**Remark.** *Theorem 1 does not impose specific assumptions on the width of the network or the initialization of the parameters. Moreover, it does not impose more assumptions on the data or labels, as long as the data lie in a one-dimensional subspace. For the rest of the paper, We focus on the case $\beta = 1$ as it admits simpler and more human-interpretable solutions (8), which will be the starting point of all following results. Some special cases of $\beta$, including $\beta = 2, 3$ are also in principle solvable, but it is difficult to write the results in a comprehensible form for a large width and we leave it to a future work to study these cases in detail. Our focus will be on the optimization dynamics of this network, even though it is also possible to discuss generalization with this solution – we briefly touch on this in Appendix D.*

Let us begin by analyzing each term and clarifying their meanings. In the theorem, we have transformed $u_i$ and $w_{ij}$ into an alternative basis $p_i$ and $q_i$, and $\xi(t)$ is the only time-dependent term. Note that $\xi$ decays exponentially towards zero at the time scale $t_c$.

The constants $\alpha_+\alpha_- = Q/P$ are two asymptotic scale factors. In the limit $t \to \infty$, we have that

$$
p_i(\infty) = p_i(0)\sqrt{\alpha_+}, \; q_i(\infty) = q_i(0)/\sqrt{\alpha_+}.
\tag{12}
$$

---

[1]Because $\alpha_+ > 0$, $\xi(t) < 1$. So, (8) is well defined. Also, See Appendix B for the case $P = 0$.

This directly gives us the mapping between the initialization to the converged solution. Unlike a strongly convex problem where the solution is independent of the initialization, we see that the converged solution for our model is strongly dependent on the initialization and on the choice of hyperparameters. Perhaps surprisingly, because $\alpha_\pm$ in (11) are functions of the learning rates, the converged solution (12) depends directly on the magnitudes of the learning rates. This directly tells us the implicit bias of gradient flow for this problem. Another special feature of the solution is that for any direction orthogonal to $x$, the model will remain unchanged during training. Let $m \perp x$, we have that $\sum_j w_{ij}(t) m_j = \sum_j w_{ij}(0) m_j$. Namely, the output of the model in the subspace where there is no data remains constant during training.

In the theorem, what is especially important is the characteristic time scale $t_c$, which is roughly the time it takes for learning to happen. Notably, the squared learning speed $t_c^{-2}$ depends on two competing factors:

$$t_c^{-2} = \underbrace{\eta_u \eta_w \gamma^2 \rho^2 y^2}_{\text{contribution from feature learning}} + \underbrace{4\rho^4(\gamma^2 d)^2 PQ}_{\text{contribution from kernel learning}}$$

The first factor depends on the input-output correlation and learning rate, which we will see is indicative of feature learning. The second term depends only on the input data and on the model initialization. We will see that when this term is dominant, the model is in the kernel regime. In fact, this result already invites a strong interpretation: the learning of the kernel regime is driven by the initialization and the input feature, whereas the learning in the feature learning regime is driven by the target mapping and large learning rates.[2]

Using this theorem, one can compute the evolution of the NTK. Note that when different learning rates are used for different layers, the NTK needs to be defined slightly differently from the conventional definition. For the MSE loss, the NTK is the quantity $K$ that enters the following dynamics: $\frac{df(x)}{dt} = 2K(x, x')(f(x') - y)$. This implies that for our problem,

$$K(x, x'; t) = \gamma^2 x^T (\eta_w W(t)^T W(t) + \eta_u \|\boldsymbol{u}(t)\|^2 I) x', \tag{13}$$

which follows from Eq. (2) and (3), where $W$ stands for the matrix with elements $w_{ij}$. $\boldsymbol{u}(t)$ and $W(t)$ are obtained via Eqs. (7) to (11). While the overall formula for the NTK dynamics is quite complex, we will provide the conditions concerning when it evolves with an $O(1)$ amount in Section 4. When $\eta_w = \eta_u$, our definition agrees with the standard NTK.

## 3.2 Learning by Alignment and Disalignment

Before we discuss the various phase diagrams implied by Theorem 1, we first focus on an interesting effect predicted by this theorem, which differentiates it from previous results on similar problems. We first set $x$ to be 1d, because Theorem 1 suggests that the dynamics of GD training has only a rank-1 effect (i.e. $\boldsymbol{p}$ and $\boldsymbol{q}$ are two effective weight vectors). Numerical results for non-1d $x$ are presented at the end of this subsection. A quantity of theoretical and practical interest is $\zeta(t) := \boldsymbol{u}^T \boldsymbol{w} / \|\boldsymbol{u}\| \|\boldsymbol{w}\|$, which represents the cosine similarity between $u$ and $w$. Recent works have identified the alignment between the weight and representation after the training starts as a mechanism for feature learning (Everett et al., 2024). Studying the evolution of $\zeta$ thus offers a direct clue of how this alignment happens. This quantity is especially interesting to study because it tells us how well-aligned the two layers are during training. Notably, this quantity vanishes as $d \to \infty$ if and only if the model is in the kernel regime (Theorem 2), so it serves as a great metric for probing how feature learning happens.

Letting $x = 1$ and denoting $\alpha(t) = \frac{\alpha_+ + \xi \alpha_-}{1 - \xi}$, we have by Theorem 1:

$$\zeta(t) = \frac{\alpha(t)P - Q/\alpha(t)}{\sqrt{(\alpha(t)P + Q/\alpha(t))^2 - (\frac{2}{d}\sum p_i q_i)^2}}, \tag{14}$$

---

[2]Another pair of important quantities are $P$ and $Q$, which are essentially the initialization scales of the model. For $\eta_u = \eta_w$, a small $P$ implies that $w_i(0) \approx -u_i(0)$ for all $i$, which implies that the model is close to anti-parallel at the start of training. Likewise, a small $Q$ implies that $w_i(0) \approx u_i(0)$ for all $i$; namely, the two layers start training when they are approximately parallel. When both $P$ and $Q$ are small, the model is initialized close to the origin, which is a saddle point. We will also see below that in the kernel regime, we always have $Q = P$.

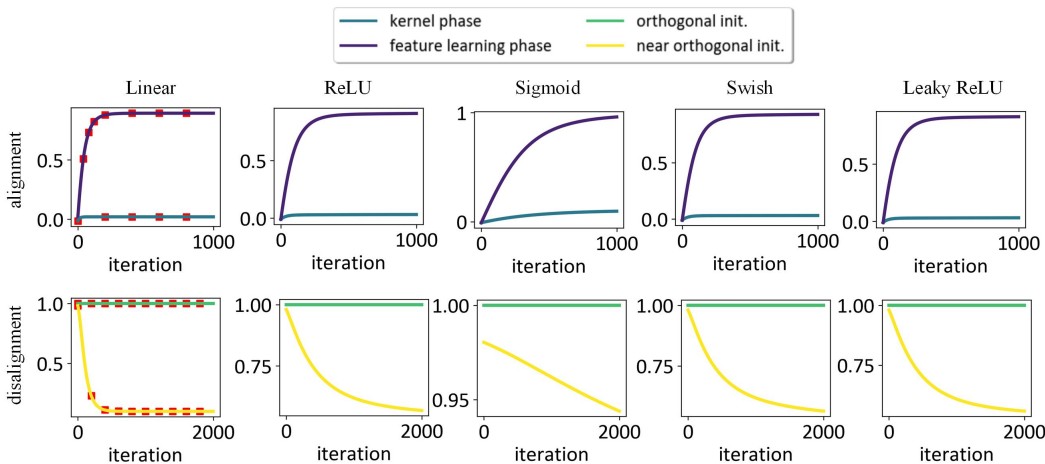

Figure 1: The evolution of $\zeta$ of two-layer networks with different settings. Specifically, we test linear, ReLU, sigmoid, swish, and leaky ReLU activations for both alignment (**upper**) and disalignment (**lower**) cases. For the linear network, we show the theoretical predictions obtained from (14) as lines and experimental results as points. The results for nonlinear networks are qualitatively similar.

where $4p_i q_i = u_i^2 - w_i^2 = const$ does not change during training. In general, the angle evolves by an $O(1)$[3] amount during training[4]. In fact, the angle remains unchanged only in the orthogonal initialization case or in the kernel phase, where $\alpha(t) = 0/1$ throughout training (see Appendix C).

For Gaussian initialization, an intriguing fact is that layers tend to align in the feature learning phase, while the alignment remains asymptotically zero for the kernel phase (see Section 4 for more formal definitions of the phases). Assuming $P \approx Q$ and $\sum_{i=1}^d p_i q_i \approx 0$, which holds for $d$ sufficiently large, (14) leads to $\zeta(t) \approx \frac{\alpha(t)^2 - 1}{\alpha(t)^2 + 1}$, which monotonously changes from 0 to $\frac{\alpha_+^2 - 1}{\alpha_+^2 + 1}$. In the feature learning phase, two terms in (10) are of the same order (see Section 4), so we can assume $\sqrt{\eta_u \eta_w} \gamma \rho y \geq 2K \rho^2 \gamma^2 dP$ without loss of generality, where $K$ is a certain positive constant. This further leads to a non-zero lower bound of the final alignment $\zeta(\infty) \geq \frac{(K+1)^2 - 1}{(K+1)^2 + 1}$. On the other hand, in the kernel phase, $\alpha_+ \approx 1$, and thus $\zeta(t) = o(1)$. Therefore, in the kernel regime, the two layers are essentially orthogonal to each other throughout training. This suggests one mechanism for the failure of the kernel learning phase. For a data point $x$, the hidden representation is $wx$, but predominantly many information in $wx$ is ignored after the the layer $u$. This implies that the model will have a disproportionately larger norm than what is actually required to fit the data, which could in turn imply strong overfitting.

When the two layers are initialized in a parallel way. This setting is often called the "orthogonal initialization" (Saxe et al., 2013). In the orthogonal initialization, $u$ is parallel to $w$, and so $p_i = Cq_i$ for a constant $C$. In this case, it is easy to verify that $\zeta(t)^2 = 1$, meaning that $u$ and $w$ remain parallel or anti-parallel throughout training.

In general, we might be interested in whether the alignment increases or decreases. Our solution implies a rather remarkable fact: $\zeta$ is always a monotonic function of $t$. To see this, its derivative is

$$\frac{d\zeta}{d\alpha} = \frac{(P + \frac{Q}{\alpha^2})(4PQ - (\frac{2}{d}\sum p_i q_i)^2)}{[(\alpha(t)P + Q/\alpha(t))^2 - (\frac{2}{d}\sum p_i q_i)^2]^{3/2}}. \tag{15}$$

When $u$ and $w$ are parallel, this quantity is zero, in agreement with our discussion about orthogonal initialization. When they are not parallel, we have that $4PQ - (\frac{2}{d}\sum p_i q_i)^2 > 0$ by the Cauchy inequality, and thus $d\zeta/d\alpha > 0$. Because $\alpha(t)$ monotonically evolves from 1 to $\alpha_+ > 0$, the evolution of $\zeta$ is also simple: $\zeta(t)$ monotonically increases if $\alpha_+ > 1$ or, equivalently, if

$$\frac{\sqrt{\eta_u \eta_w} y}{2\gamma d\rho P} + \sqrt{\left(\frac{\sqrt{\eta_u \eta_w} y}{2\gamma d\rho P}\right)^2 + \frac{Q}{P}} > 1 \tag{16}$$

and monotonically decreases if $\alpha_+ < 1$. $\zeta$ does not change if $\alpha_+ = 1$.

---

[3]It is because $\alpha(t)$ changes from 1 to $\alpha_+$ during training. In the feature learning phase, $\alpha_\pm = O(1)$, so the angle evolves by O(1). See the proof of Theorem 2 for more details.

[4]$f(x) = O(g(x))$ means that$f(x) = O(g(x))$ holds almost surely.

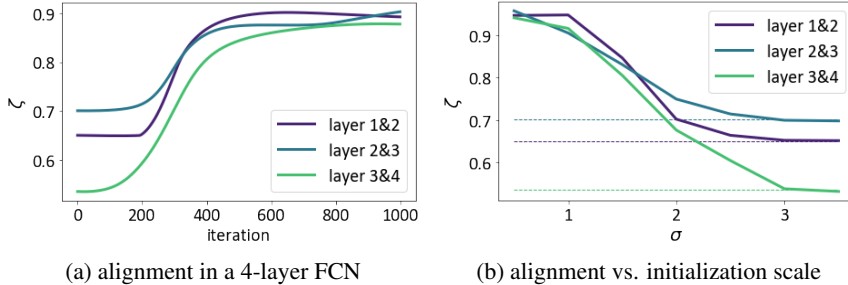

(a) alignment in a 4-layer FCN        (b) alignment vs. initialization scale

Figure 2: The alignment angle $\zeta$ between different layers of a four-layer FCN with ReLU activation trained on MNIST. (b) shows the final alignment for different initialization scale $\sigma$, while (a) shows training curves corresponding to $\sigma = 1$. The dashed lines in (b) show the initial alignment. See Appendix C for experiments on a six-layer network.

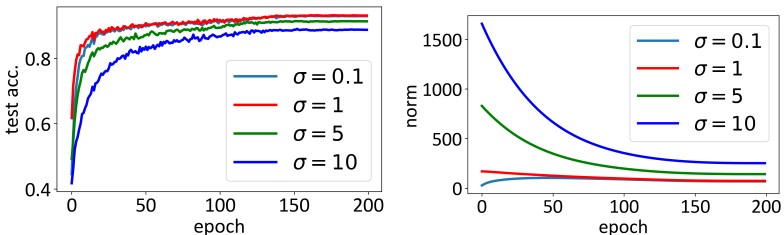

Figure 3: The initialization scale $\sigma$ correlates negatively with the performance of Resnet-18 on the CIFAR-10 dataset. **Left**: test accuracy. Here, $\sigma$ is a constant multiplier we apply to the initialized weights of the model under the Kaiming init. **Right**: the norm of all weights. While all models achieve a 100% training accuracy, models initialized with a larger scale converge to solutions with higher weight norms, which is a sign that the layers are misaligned.

When does condition (16) hold? Let us focus on the case $y > 0$ because the theory is symmetric in the sign of $y$. The first observation is that it holds whenever $Q \geq P$, which is equivalent to $u^T(0)w(0) < 0$. Namely, if the model is making wrong predictions from the beginning, it will learn by aligning different layers. Moreover, this quantity also depends on the balance of the two learning rates. Notably, when the learning rates for the two different layers are the same, the change in $\zeta$ is independent of the learning rate. The dependence on the learning rate becomes significant once we use different learning rates on the two layers. For example, when $\eta_u \gg \eta_w$ (or vice versa), this condition depends monotonically and (essentially) linearly on $\eta_w$, and making $\eta_w$ close to $\eta_u$ has the effect of making the two layers more aligned.

Why does the alignment effect depend on the ratio $Q/P$? Because when $Q$ is small, the model layers are initialized to be aligned and are likely to make predictions that are too large, and the learning process necessarily involves decreasing the model output on the data points, which can be achieved in one of the two ways: (1) decrease the scale $\|u\|\|w\|$, or (2) decrease the alignment $\zeta$. When condition (16) is not satisfied, GD employs both mechanisms for learning. Lastly, it is also worth noting that for this problem, SGD has been shown to converge to a perfectly aligned distribution of solutions (Ziyin et al., 2023). This comparison thus shows a qualitative difference between GD and SGD – using GD, alignment is a strong function of the initialization, whereas in SGD, the alignment is quite independent of the initialization. The difference between SGD and GD is of order 1 in this problem, even if the noise is very small.

See Figure 1, where we show that the evolution of $\zeta$ of two-layer networks with $d = 10000$. It is trained on a regression task. Similar experimental results are observed for a classification task trained with the cross-entropy loss (Appendix C). We choose $\gamma = 1/\sqrt{d}$ for the kernel phase and $\gamma = 1/d$ for the feature learning phase. The initial weights are sampled from i.i.d. Gaussian distribution $\mathcal{N}(0, 1)$. Therefore, we have $P \approx Q$ and the initial $\zeta(0) \approx 0$. From (11) we have $\alpha_+ > 1$ if $y > 0$, so $\zeta(t)$ monotonically increases, but the increase is negligible in the kernel phase. Therefore, in this case, the model learns features by alignment. Meanwhile, the orthogonal initialization remains 1 as predicted. In the near orthogonal case of Figure 1, we set $\zeta(0) \approx 1$ and the initial model output to be large. As predicted, $\zeta(t)$ monotonically decreases. In this case, the model learns by disalignment. From Figure 1, we also see that this phenomenon holds for all non-linear activation functions.

The layer alignment and disalignment effects can be generalized to higher dimensions, deeper networks and non-linear activations. Here, we define $\zeta := \frac{\|UW\|}{\|U\|\|W\|}$, where $U, W$ are the weight matrices of two consecutive layers and the L2 norm for matrices is used. See Figure 2 for a four-layer fully connected network (FCN) with ReLU activation and different initialization scales trained on MNIST datasets. Figure 2 (a) shows that the alignment between consecutive layers increases during training, and Figure 2 (b) demonstrates that layers stop being aligned for large initialization. These results are consistent with simpler settings, verifying the generality of the discovered mechanism.

## 3.3 Learning by Rescaling

Learning can also happen by rescaling the output. The evolution of $\|\boldsymbol{u}\|$ and $\|\boldsymbol{w}\|$ are given by

$$\|\boldsymbol{u}\|^2 = d(\alpha_+ P + \frac{Q}{\alpha_+}) - 2\sum_{i=1}^d p_i q_i, \quad \|w\|^2 = d(\alpha_+ P + \frac{Q}{\alpha_+}) + 2\sum_{i=1}^d p_i q_i,$$

and, thus, $\frac{d\|\boldsymbol{u}\|^2}{d\alpha_+} = \frac{d\|\boldsymbol{w}\|^2}{d\alpha_+} = d(P - \frac{Q}{\alpha_+^2})$, which is positive when $\zeta > 0$, and negative when $\zeta < 0$. Thus, the rescaling coincides with the alignment, namely, $\|\boldsymbol{u}\|$ and $\|\boldsymbol{w}\|$ become larger when they are being aligned ($|\zeta|$ gets larger), and become smaller when they are being disaligned ($|\zeta|$ gets smaller). More explicitly, (1) $P > Q$ and $\alpha_+ > 1$, or $P < Q$ and $\alpha_+ < 1$, or $P = Q$: $\|\boldsymbol{u}\|$ and $\|\boldsymbol{w}\|$ monotonically increase. (2) $P > Q$ and $1 > \alpha_+ \geq \sqrt{Q/P}$, or $P < Q$ and $1 < \alpha_+ \leq \sqrt{Q/P}$: $\|\boldsymbol{u}\|$ and $\|\boldsymbol{w}\|$ and monotonically decrease. (3) $P > Q$ and $\alpha_+ < \sqrt{Q/P}$, or $P < Q$ and $\alpha_+ > \sqrt{Q/P}$: $\|\boldsymbol{u}\|$ and $\|\boldsymbol{w}\|$ first decrease, and then increase. (4) $\alpha_+ = 1$: everything keeps unchanged. Again, in the kernel phase, the scale change of the model vanishes. In the orthogonal initialization, however, this quantity changes by an $O(1)$ amount. Therefore, the orthogonal initialization essentially learns by rescaling the magnitude of the output.

## 3.4 How does feature learning happen?

The analysis thus suggests three mechanisms for feature learning, all of which are absent in the kernel phase. The first two mechanisms are the alignment and disalignment in the hidden layer, which is driven by the initialization balancing between the two layers. The second mechanism is the rescaling output, which is a simple operation and is unlikely to be related to learning actual features. This argument also agrees with the common technique that even if we normalize the layer output, the performance of the network does not deteriorate (Ioffe & Szegedy, 2015).

The second question is whether we want alignment or disalignment. The intuitive answer seems to be that alignment should be preferred over disalignment. Because aligned layers require a smaller model norm to make the same prediction, whereas a disaligned model requires a very large model norm to make the prediction. Our theory thus offers a mechansism of how relatively smaller initialization is often more preferable in deep learning – when the model has an overly large initialization, it will learn by disalignment, whereas a small initialization prefers alignment. This is in agreement with the common observation that a larger initialization variance correlates strongly with a worse performance (Sutskever et al., 2013; Xu et al., 2019; Zhang et al., 2020). The parameterization in Yang et al. (2022) ensures that activations are O(1) at initialization, which could avoid disalignment problems. This is distinct from the NTK/feature-learning explanation, which suggests that larger initializations push the model towards the kernel regime, as discussed in Section 4.2 of our paper. In the kernel regime, alignment and disalignment effects are not present. Thus, our explanation is particularly relevant for cases where the initialization is large but not large enough to push the model into the kernel regime. While our explanation complements existing theories, it provides a distinct angle on the role of initialization in training dynamics.

A numerical result is presented in Figure 3, where a larger initialization leads to worse performance. Note that this example can only be explained through the disalignment effect because (1) the model achieves 100% train accuracy in all settings, yet (2) a larger initialization leads to a larger norm at the end of the training, which also correlates with worse performance. Another piece of evidence is the commonly observed underperformance of kernel models. In the kernel phase, the model norm diverges and the model alignment is always zero, which could be a hint of strong overfitting. Therefore, our theory suggests that it would be a great idea for future works to develop algorithms that maximize layer alignment while minimizing the change in the output scale.

## 4 PHASE DIAGRAMS

Our theory can be applied to study the learning of different scaling limits, where we scale the hyperparameters with a scaling parameter $\kappa \to \infty$. Here, $\kappa$ is an abstract quantity that increases linearly, and all the hyperparameters including the width are a power-law function of $\kappa$. Conventionally, $\kappa$ is the model width; however, this excludes the discussion of the lazy training regime in the theory, where the model width is kept fixed and the scaling parameter is the model output scale $\gamma$.

We first establish the necessary and sufficient condition for learning to happen: the learning time $t_c$ needs to be of order $\Theta(1)$. When it diverges, learning is frozen at initialization. When it vanishes to zero, the discrete-time SGD algorithm will be unstable, a point that is first pointed out by Yang & Hu (2020). Therefore, we first study the condition for $t_c$ to be of order 1, which is equivalent to the condition that (assuming $x, y$ are order 1)

$$\eta_u \eta_w \gamma^2 + (\gamma^2 d)^2 PQ = \Theta(1). \quad (17)$$

For Gaussian initialization $u_{i0} \sim \mathcal{N}(0, \sigma_u^2)$ and $w_{i0} \sim \mathcal{N}(0, \sigma_w^2)$, $P$ and $Q$ are random variables with expectation $(\eta_w \sigma_u^2 + \eta_u \sigma_w^2)/4$ and variance $(\eta_w \sigma_u^2 + \eta_u \sigma_w^2)^2/8d$. Generally, all hyperparameters are powers of $\kappa$: $d \propto \kappa^{c_d}$, $\gamma \propto \kappa^{c_\gamma}$, $\sigma_w^2 \propto \kappa^{c_w}$, $\sigma_u^2 \propto \kappa^{c_u}$, $\eta_w \propto \kappa^{c_{\eta_w}}$ and $\eta_u \propto \kappa^{c_{\eta_u}}$[5]. For simplicity, we set the input dimension $d_0$ to be a constant.

Table 1: Phases of learning in different scaling limits. For brevity, the learning rates of the two layers are set to be equal. The first block shows that the models can be frozen or unstable if we do not scale $\eta$ accordingly. The second block shows that one can always choose $\eta$ such that the model training is stable and does not freeze. The third and fourth blocks show that one can always choose a pair of $\eta$ and $\gamma$ such that the model is either in the feature learning phase or the kernel phase. MF refers to the mean-field scaling in (Mei et al., 2018) and lazy refers to the scaling in (Chizat et al., 2018).

| scaling | NTK | MF | Xavier | Kaiming | lazy |
|---|---|---|---|---|---|
| $c_d$ | 1 | 1 | 1 | 1 | 0 |
| $c_\gamma$ | -1/2 | -1 | 0 | 0 | 1 |
| $c_u$ | 0 | 0 | -1 | -1 | 0 |
| $c_w$ | 0 | 0 | -1 | 0 | 0 |
| $c_\eta$ | 0 | 0 | 0 | 0 | 0 |
| phase | kernel | frozen | learning | unstable | unstable |
| $c_\eta^*$ | 0 | 1 | 0 | -1 | -2 |
| phase | kernel | learning | learning | kernel | kernel |
| $c_\eta^+$ | 1 | 1 | 0 | 1 | 0 |
| $c_\gamma^+$ | -1 | -1 | 0 | -1 | 0 |
| phase | learning | learning | learning | learning | learning |
| $c_\eta^-$ | 0 | 0 | -2 | -1 | -2 |
| $c_\gamma^-$ | -1/2 | -1/2 | 1 | 0 | 1 |
| phase | kernel | kernel | kernel | kernel | kernel |

Equation (17) implies

$$\max\{2c_\gamma + c_{\eta_u} + c_{\eta_w}, \ 2c_\gamma + c_d + \max\{c_{\eta_w} + c_u, c_{\eta_u} + c_w\}\} = 0. \quad (18)$$

Whatever choice of the exponents that solves the above equation is a valid learning limit for a neural network. The phase of the network depends on the relative order of the above two terms.

**Definition 1.** *A model is in the kernel phase if (1) Eq. (13) is independent of $t$ as $\kappa \to \infty$ (2) $NTK = \Theta(1)$[6].*

When $t_c = \Theta(1)$, a model is said to be in the feature learning phase if it is not in the kernel phase.

**Theorem 2.** *When Eq. (18) holds, a model is in the kernel phase if and only if $\lim_{\kappa \to \infty} P/Q = 1$, a.s.. and*

$$c_d + \max\{c_{\eta_w} + c_u, c_{\eta_u} + c_w\} > c_{\eta_u} + c_{\eta_w}. \quad (19)$$

The necessary condition $P \approx Q$ for the model being in the kernel phase is interesting and highlights the important role of initialization in deep learning. There are three common cases when this holds:

1. $d \to \infty$ and $u_0$ and $w_0$ are independent (standard NTK);
2. $d$ is finite and the initial model output is zero: $\sum_{i=1}^{d} \sum_{j=1}^{d_0} u_i w_{ij} x_j = 0$ (lazy training)
3. $d$ is finite, $\kappa \to \infty$ and $c_u + c_{\eta_w} \neq c_w + c_{\eta_u}$;

The first case is the standard way of initialization, from which one can derive the classic analysis of the kernel phase by invoking the law of large numbers. The second case is the assumption used in the lazy training regime (Chizat et al., 2018). (Chizat et al. (2018) assumes $c_\gamma = 1$, $c_{\eta_u} = c_{\eta_w} = -2$ and $c_u = c_w = 0$, satisfying the conditions of the exponents (18) and (19).) This case, however, relies on a special initialization, and thus our results better illustrate the occurrence of the kernel

---

[5]Note that the solution in Theorem 1 is invariant under the transform $c_u \to c_u + \theta, c_w \to c_w + \theta, c_\gamma \to c_\gamma - 2\theta, c_{\eta_u} \to c_{\eta_u} + 2\theta, c_{\eta_w} \to c_{\eta_w} + 2\theta$, corresponding to the abc symmetries in Yang & Hu (2020).

[6]This requires $\gamma^2(\eta_w W^T W + \eta \|\boldsymbol{u}\|^2 I) = \Theta(1)$.

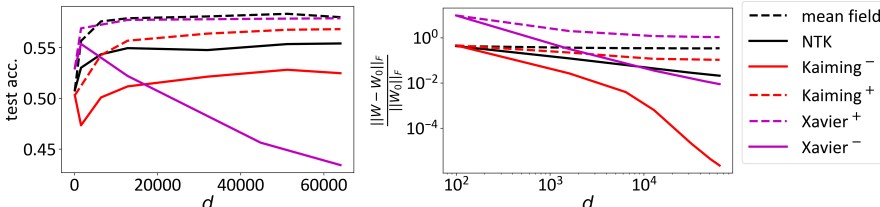

Figure 4: A two-layer fully connected ReLU net with $d$ neurons trained on the CIFAR-10 dataset for $10000$ epochs with batch size 128. The kernel phase is shown in solid lines and the feature learning phase is shown in dashed lines. As the theory predicts, both types of initialization can be turned into either the feature learning or the kernel phase by choosing different combinations of $\gamma$ and $\eta$. **Left**: the best test accuracy during training. **Right**: relative distance from the initialization.

phase for advanced initialization methods where different weights can be correlated. The third case happens when the learning rate and the initialization are not balanced. This suggests that to achieve feature learning, one should make sure that the learning rate and the initialization are well balanced: $c_u = c_w$.

In conclusion, the overall phase is (1) **kernel phase**, if the first term in (18) is strictly smaller than the second term: $0 = 2c_\gamma + c_d + \max\{c_{\eta_w} + c_u, c_{\eta_u} + c_w\} > 2c_\gamma + c_{\eta_u} + c_{\eta_w}$ and $\lim_{\kappa \to \infty} P/Q = 1$, (2) **feature learning phase** if otherwise. A key difference between these two phases is whether the evolution of the NTK is $O(1)$, or equivalently whether the model learns features.

The following corollaries are direct consequences of Eq. (18).

**Corollary 1.** *For any $c_\gamma$, $c_u$ and $c_w$, choosing $c_{\eta_u} = c_{\eta_w} = \min\{-c_\gamma, -2c_\gamma - c_d - \max\{c_u, c_w\}\}$ ensures that the model is stable.*

**Corollary 2.** *For any $c_u$ and $c_w$, choosing $c_{\eta_u} = c_{\eta_w} = c_\eta$ and $c_\gamma = -c_\eta$ with $c_\eta \geq c_d + \max\{c_u, c_w\}$ leads to a feature learning phase.*

**Corollary 3.** *Assume $\lim_{\kappa \to \infty} P/Q = 1$. For any $c_u$ and $c_w$, choosing $c_\gamma = -\frac{1}{2}(c_d + \max\{c_u, c_w\} + c_\eta)$ and $c_\eta < c_d + \max\{c_u, c_w\}$ leads to a kernel phase.*

They imply two important messages: for every initialization scheme, (1) one can choose an optimal learning rate such that the learning is stable; (2) one can choose an optimal pair of learning rate and output scale $\gamma$ such that the model is in the feature learning phase. Point (1) agrees with the analysis in Yang & Hu (2020), whereas point (2) is a new insight we offer. See Table 1 for the classification of different common scalings. We choose scalings according to Corollary 2 and 3, to turn each model into the feature learning or the kernel phase. Table 1 is closely related to the Tensor programs framework (Yang & Hu, 2020; Yang et al., 2022; Yang & Littwin, 2023). The key difference is that our results apply to finite-width networks with arbitrary initialization, whereas Tensor programs assume infinite width and Gaussian initialization.

## 4.1 PHASES DIAGRAM OF INFINITE-WIDTH MODELS

Now, let us focus on the case $\kappa = d \to \infty$ ($c_d = 1$), corresponding to the infinite width limit in the NTK and feature learning literature (Jacot et al., 2018; Li et al., 2020; Yang & Hu, 2020).

See Figure 4. We implement a two-layer FCN on the CIFAR-10 dataset with ReLU activation. We run experiments with the scalings of the standard NTK, standard mean-field, Kaiming model, and Xavier model. $c_\gamma$ and $c_\eta$ are chosen according to Table 1. Here, we use the superscript + to denote the type of scaling that leads to a feature learning phase, and − denotes the kernel phase. For the Kaiming and Xavier model, we choose both $c_\eta^\pm$ and $c_\gamma^\pm$, and refer them as Kaiming$^\pm$ and Xavier$^\pm$, respectively. The left figure shows that turning the Kaiming model into the feature learning phase improves the test accuracy by approximately $5\%$, similar to the gap between the standard NTK model and the mean-field model. Meanwhile, turning the Xavier model into the kernel phase decreases the test accuracy by approximately $10\%$. This is because the fixed kernel restricts the generalization ability in the kernel phase, and the difference between these models in the kernel phase might be attributed to their different kernels. The right figure asserts that there is a power law scaling of the weight evolution $\frac{\|W - W_0\|}{W_0} \propto d^{-\delta}$, and we can predict that $\delta = 0$ for the feature learning phase, and $\delta = 0.5, 1, 2$ for NTK, Xavier$^-$ and Kaiming$^-$, respectively, which are perfectly consistent with the numerical results ($\delta = 0.47, 1.08, 1.93$). See Appendix B.4 for details.

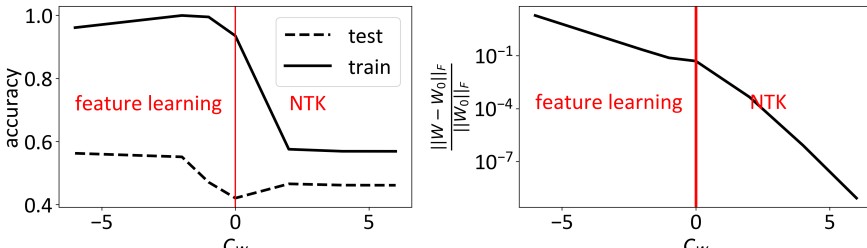

Figure 5: A two-layer FCN with different initialization scales trained on the CIFAR-10 dataset. We see that finite-width models can also exhibit qualitative differences between the feature learning and the kernel phases when other hyperparameters are scaled toward infinity. Notably, this scaling is different from the lazy training scaling, implying that there are numerous (actually infinitely many) ways for the model to enter the kernel phase, even at a finite width.

Thus, in agreement with the theory, choosing different combinations of the output scale $\gamma$ and $\eta$ can turn any initialization into the feature learning phase. This insight could be very useful in practice, as the Kaiming init. is predominantly used in deep learning practice and is often observed to have better performance at common widths of the network. Our result thus suggests that it is possible to keep its advantage even if we scale up the network. Further, our results also imply that any valid learning regimes transfer well when the model gets larger, while in the feature learning phase, a larger model generally leads to better performance. This is consistent with Yang et al. (2022).

## 4.2 PHASE DIAGRAM FOR INITIALIZATIONS

Now, we study the case when $d$ is kept fixed ($c_d = 0$), while other variables scale with $\kappa \to \infty$. In this case, the phase diagram is also given by Theorem 2. See Figure 5 for an experiment. We set $c_u = \max\{0, c_w\}$ and $c_\gamma = \min\{-c_w/2, 0\}$. This choice satisfies (18). By Theorem 2, the network is in the kernel phase if and only if $c_w > 0$.

One important example for this section is the lazy training regime, where $c_u = c_w = c_d = 0$, and we can choose $c_\gamma = 1$ and $c_\eta = -2$ according to Corollary 3, leading to a kernel phase in finite width. Another example is to consider large initialization, i.e., $c_u = c_w = c > 0$. In this case, we can choose $c_\eta = c$ and $c_\gamma = -c$ according to Corollary 2, leading to a feature learning phase. Actually, this choice of the normalization factor $\gamma$ cancels out the scaling of the initialization. On the other hand, if we choose $\gamma = 1$ as commonly done, we have to choose $c_\eta = -c$ according to Corollary 3, leading to a kernel space. This might be another possible explanation that larger initialization often leads to worse performance empirically. Like before, we implement a two-layer fully connected ReLU network on the CIFAR-10 dataset with $d = 2000$. We choose $\kappa = 10$ for illustration purposes. A clear distinction is observed between the feature learning phase and the kernel phase. (1) In Figure 5(a), the training accuracy can reach 1.0 in the feature learning phase but not the kernel phase, because the NTK in the kernel phase is fixed, and thus the best training accuracy is limited by the fixed kernel. (2) As discussed in the previous section, the test accuracy In Figure 5(a) is about 5% higher in the feature learning phase due to its trainable kernel. (3) In Figure 5(b), the weight matrices evolve significantly in the feature learning phase but not the kernel phase.

## 5 CONCLUSION

Solving minimal models has been a primary approach in natural sciences to understand how controllable parameters are causally related to phenomena. In this work, we have solved the learning dynamics of a minimal finite-width model of a two-layer linear network. Through a comprehensive analysis of its learning dynamics and phase diagrams, we have uncovered valuable insights into how feature learning happens and the impact of various scalings on the training dynamics of nonlinear neural networks. Our theory is obviously limited: the analytical results only hold for inputs lying in a one-dimensional subspace. This limitation arises from the inadequacy of conservation laws in more general cases (e.g., Marcotte et al. (2023, Corollary 4.4) suggests that the maximal number of independent conservation laws might be much less than the degrees of freedoms), which potentially implies the impossibility of solving a more general model than ours. Lastly, our results only considers a deterministic learning dynamics; feature learning actual models are likely to be also determined by regularization and noise during training (Ziyin et al., 2025a;b), and it could be interesting to compare feature learning under noise and without noise.

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

# A    ADDITIONAL RELATED WORK

There are also other recent studies focusing on feature learning beyond the kernel regime. Li et al. (2020) demonstrates that two-layer networks outperform kernel methods, while Damian et al. (2022); Abbe et al. (2022) highlight functions learnable by gradient descent in two-layer networks but not by kernel methods. Moreover, several studies (Ba et al., 2022; Dandi et al., 2023; Cui et al., 2024) show that a few gradient steps help neural networks adapt to dataset features, improving generalization. Our work aims to provide new insights into feature learning through an analytically solvable model.

Another critical regime for feature learning involves large learning rates. Jastrzebski et al. (2020); Long (2021); Lewkowycz et al. (2020); Wang et al. (2024) reveal that large learning rates offer benefits such as better conditioning of kernel and Hessian matrices and improved generalization. Notably, while the catapult (uv) model (Lewkowycz et al., 2020) shares some similarity with our model, their focus is completely different. Lewkowycz et al. (2020) uses the NTK scaling, and shows that neural networks can escape the kernel regime with sufficiently large learning rates. On the other hand, we show that how different scalings influence the phases of neural networks with sufficiently small learning rates. Therefore, the mechanisms of feature learning proposed in this paper do not appear in Lewkowycz et al. (2020). Moreover, although Kalra et al. (2023); Kalra & Barkeshli (2024) try to extend the catapult (uv) model to finite-width settings with different parameterizations, their results remain qualitative, perturbative, or focus solely on fixed points without deriving a complete analytic solution. In contrast, our work provides a fully solvable model with exact dynamics, even for finite widths, under more general initialization and parameterization settings. It is a promising future work to extend our gradient flow solutions to finite learning rates and adaptive optimizers, which can influence the regime (Yang & Littwin, 2023; Wang et al., 2024).

The concept of "alignment" is explored in many works. For instance, Lewkowycz et al. (2020) discusses alignment between feedforward activations and backpropagated gradients in the catapult mechanism, a feature of the large learning rate regime, absent in our model. Seroussi et al. (2023) investigates alignment between backpropagation and weight matrices in neural network Gaussian processes. Some studies (Baratin et al., 2021; Atanasov et al., 2021; Loo et al., 2022; Wang et al., 2024) examine alignment effects between models and data. Our definition of alignment differs from them. However, we note that our definition of alignment is inherently equivalent with the "alignment ratio" measured in a recent work (Everett et al., 2024), which focuses on how alignment across layers influences model scaling and supports the importance of analyzing how alignment changes along training.

Finally, while many studies analyze the convergence (Nguegnang et al., 2021; Bietti et al., 2023) or dynamics (Jacot et al., 2021; Arous et al., 2021; Paquette et al., 2021) of gradient descent without exact solutions, they typically prove limited properties or asymptotic results. We believe an exactly solvable model offers a deeper understanding of gradient descent dynamics.

# B    THEORETICAL CONCERNS

## B.1    PROOF OF PROPOSITION 1

*Proof.* To make the analysis more concrete, we consider the standard loss function

$$\tilde{L}(u, w) = \frac{1}{N} \sum_{k=1}^{N} (\gamma \sum_{i=1}^{d} u_i (\sum_{j=1}^{d_0} w_{ij} \tilde{x}_{jk})^\beta - \tilde{y}_k)^2, \tag{20}$$

where $N$ is the size of the training set.

Data points lie in a 1d-subspace, meaning that $\tilde{x}_{jk} = a_k n_j$ for a constant unit vector $\boldsymbol{n}$. Due to the 1d nature of the data, the training dynamics on this loss function is completely identical to training on the following loss $L(u, w) = (\gamma \sum_{i=1}^{d} u_i (\sum_{j=1}^{d_0} w_{ij} x_j)^\beta - y)$, where $x_j = \sqrt{\sum_{k=1}^{N} a_k^{2\beta}} n_j$ and

Table 2: Varibles used for Theorem 1.

| variables | explanation |
|---|---|
| $\boldsymbol{u} \in \mathbb{R}^d, W \in \mathbb{R}^{d \times d_0}$ | weight vector/matrix |
| $\boldsymbol{x} \in \mathbb{R}^{d_0}, y \in \mathbb{R}, \rho \coloneqq \sqrt{\frac{1}{d_0} \sum_{i=0}^{d_0} x_i^2}$ | effective data point and signal strength |
| $\eta_u, \eta_w, \gamma$ | learning rates and normalization factor |
| $\begin{cases} p_i(t) \coloneqq \frac{1}{2\rho}(\sqrt{\eta_u} \sum_{j=1}^{d_0} w_{ij}(t)x_j + \sqrt{\beta\eta_w}\rho u_i(t)) \\ q_i(t) \coloneqq \frac{1}{2\rho}(\sqrt{\eta_u} \sum_{j=1}^{d_0} w_{ij}(t)x_j - \sqrt{\beta\eta_w}\rho u_i(t)) \end{cases}$ | generalized coordinates |
| $P \coloneqq \frac{1}{d} \sum_{j=1}^d p_j(0)^2, \ Q \coloneqq \frac{1}{d} \sum_{j=1}^d q_j(0)^2$ | sufficient statistics of generalized coordinates |
| $t_c \coloneqq 1/\left(\sqrt{\eta_u \eta_w \gamma^2 \rho^2 y^2 + 4\rho^4 (\gamma^2 d)^2 PQ}\right)$ | characteristic learning time |
| $\alpha_\pm \coloneqq \frac{1}{2(\gamma^2 d)\rho^2 P}\left(\sqrt{\eta_u \eta_w}\gamma\rho y \pm t_c^{-1}\right)$ | characteristic learning scale |
| $\xi(t) \coloneqq \frac{1-\alpha_+}{1+\alpha_-} \exp\left(-4t/t_c\right)$ | characteristic learning curve |

$y = \frac{\sum_{k=1}^N a_k^\beta \tilde{y}_k}{\sqrt{\sum_{k=1}^N a_k^{2\beta}}}$. This is because

$$
\begin{aligned}
\tilde{L}(u, w) &= (\sum_{k=1}^N a_k^{2\beta})(\gamma \sum_{i=1}^d u_i (\sum_{j=1}^{d_0} w_{ij} n_j)^\beta)^2 - 2(\sum_{k=1}^N a_k^\beta \tilde{y}_k)(\gamma \sum_{i=1}^d u_i (\sum_{j=1}^{d_0} w_{ij} n_j)^\beta) + \sum_{k=1}^N y_k^2 \\
&= L(u, w) + \sum_{k=1}^N y_k^2 - y^2.
\end{aligned}
\tag{21}
$$

Therefore, without loss of generality, the training on the standard loss $\tilde{L}$ is identical to the training on $L$ because the difference is only by a constant that does not affect gradient descent training. This setting is thus equivalent to the case when the dataset contains only a single data point $(x, y)$.[7] As is clearly shown from this example, using the notation in terms of $x$ and $y$ is much simpler to understand than using $\tilde{x}_{jk}$ and $y_k$. We believe that this notation is necessary and greatly facilitates the later discussions once the readers accept it.

Finally, all these notations can also be written in terms of $\mathbb{E}_x \coloneqq \frac{1}{N} \sum_{k=1}^N$, which is the notation we chose for introducing the lemma. $\qquad\square$

### B.2 PROOF OF THEOREM 1

*Proof.* We summarize the definitions of all variables in Table 2.

To begin with, the gradient flow reads

$$
\begin{aligned}
\frac{du_i}{dt} &= -\eta_u \frac{\partial L}{\partial u_i} = -2\eta_u \gamma (\sum_{j=1}^{d_0} w_{ij} x_j)^\beta \left(\gamma \sum_{i=1}^d u_i (\sum_{j=1}^{d_0} w_{ij} x_j)^\beta - y\right), \\
\frac{dw_{ij}}{dt} &= -\eta_w \frac{\partial L}{\partial w_{ij}} = -2\beta\eta_w \gamma u_i (\sum_{j=1}^{d_0} w_{ij} x_j)^{\beta-1} x_j \left(\gamma \sum_{i=1}^d u_i (\sum_{j=1}^{d_0} w_{ij} x_j)^\beta - y\right),
\end{aligned}
\tag{22}
$$

which implies the following two conservation laws

$$
\frac{d}{dt}(\eta_u \sum_{j=1}^{d_0} w_{ij}^2 - \beta\eta_w u_i^2) = 0,
\tag{23}
$$

$$
\frac{d}{dt}\left(\frac{w_{ij}}{x_j} - \frac{w_{ij'}}{x_{j'}}\right) = \frac{1}{x_j}\frac{dw_{ij}}{dt} - \frac{1}{x_{j'}}\frac{dw_{ij'}}{dt} = 0.
\tag{24}
$$

---

[7]Essentially, this is because we only need two points to specify a line. Also, it is trivial to extend to the case when $\tilde{y}$ is a vector that spans only a one-dimensional subspace.

From Eq. (22), we can denote $\frac{dw_{ij}}{dt} = A_i x_j$, which leads to

$$
\begin{aligned}
\frac{d}{dt} & \left( \sum_{j=1}^{d_0} w_{ij}^2 - \frac{1}{\sum_{j=1}^{d_0} x_j^2} (\sum_{j=1}^{N} w_{ij} x_j)^2 \right) \\
& = A_i \left( 2 \sum_{j=1}^{d_0} w_{ij} x_j - 2 \frac{\sum_{j=1}^{d_0} w_{ij} x_j}{\sum_{j=1}^{d_0} x_j^2} \sum_{j=1}^{d_0} x_j^2 \right) = 0.
\end{aligned}
\tag{25}
$$

According to the definitions $p_i(t) := \frac{1}{2\rho}(\sqrt{\eta_u} \sum_{j=1}^{d_0} w_{ij}(t) x_j + \sqrt{\beta \eta_w} \rho u_i(t))$ and $q_i(t) := \frac{1}{2\rho}(\sqrt{\eta_u} \sum_{j=1}^{d_0} w_{ij}(t) x_j - \sqrt{\beta \eta_w} \rho u_i(t))$, we have

$$
\begin{aligned}
\frac{d}{dt} (p_i(t) q_i(t)) & = \frac{1}{4} \frac{d}{dt} \left( \frac{\eta_u}{\sum_{j=1}^{d_0} x_j^2} (\sum_{j=1}^{N} w_{ij} x_j)^2 - \beta \eta_w u_i^2 \right) \\
& = \frac{1}{4} \frac{d}{dt} \left( \eta_u \sum_{j=1}^{d_0} w_{ij}^2 - \beta \eta_w u_i^2 \right) = 0
\end{aligned}
\tag{26}
$$

Further, substituting (22) into the definition of $p_i$ and $q_i$, we have

$$
\frac{dp_i}{dt} = -2\gamma \sqrt{\beta \eta_u \eta_w} p_i \rho \left( \frac{(p_i + q_i)\rho}{\sqrt{\eta_u}} \right)^{\beta-1} \left( \sum_{j=1}^{d} \gamma \frac{p_j - q_j}{\sqrt{\beta \eta_w}} \left( \frac{(p_j + q_j)\rho}{\sqrt{\eta_u}} \right)^{\beta} - y \right).
\tag{27}
$$

If we denote $c_i := p_i(t) q_i(t)$, we have

$$
\frac{1}{p_i(p_i + c_i/p_i)^{\beta-1}} \frac{dp_i}{dt} = \frac{1}{p_j(p_j + c_j/p_j)^{\beta-1}} \frac{dp_j}{dt},
\tag{28}
$$

which gives

$$
F_i(p_i(t)) - F_i(p_i(0)) = F_j(p_j(t)) - F_j(p_j(0)).
\tag{29}
$$

where

$$
F_i(x) := \int \frac{dx}{x(x + c_i/x)^{\beta-1}}.
\tag{30}
$$

Therefore, (27) reduces to a differential equation of $p_i$

$$
\begin{aligned}
\frac{dp_i(t)}{dt} = & -2\gamma \sqrt{\beta \eta_u \eta_w} p_i(t) \rho \left( \frac{(p_i(t) + c_i/p_i(t))\rho}{\sqrt{\eta_u}} \right)^{\beta-1} \\
& \left( \sum_{j=1}^{d} \gamma \frac{p_j(t) - c_j/p_j(t)}{\sqrt{\beta \eta_w}} \left( \frac{(p_j(t) + c_j/p_j(t))\rho}{\sqrt{\eta_u}} \right)^{\beta} - y \right).
\end{aligned}
\tag{31}
$$

Now we denote $\Delta_i(t) := F_i(p_i(t))$. Then we have

$$
\frac{d\Delta_i(t)}{dt} = -2\gamma \sqrt{\beta \eta_u^{2-\beta} \eta_w} \rho^{\beta} \left( \gamma \rho^{\beta} (\beta \eta_u^{\beta} \eta_w)^{-1/2} \sum_{j=1}^{d} (p_j(t) - c_j/p_j(t))(p_j(t) + c_j/p_j(t))^{\beta} - y \right),
\tag{32}
$$

where

$$
p_j(t) = F_j^{-1}(\Delta_i(t) - \Delta_i(0) + \Delta_j(0)).
\tag{33}
$$

(32) is an ODE with only one unknown function $\Delta_i(t)$. However, it is in general not possible to solve (32), and this is why we only focus on $\beta = 1$.

For the special case $\beta = 1$, (31) reduces to

$$
\frac{dp_i(t)}{dt} = -2\gamma \sqrt{\eta_u \eta_w} p_i(t) \rho \left( \sum_{j=1}^{d} (p_j(t)^2 - q_j(t)^2) \frac{\gamma \rho}{\sqrt{\eta_u \eta_w}} - y \right)
\tag{34}
$$

For $\beta = 1$, we also have $F_i(x) = \log x$ with its inverse $F_i^{-1}(x) = e^x$, and thus (33) reduces to

$$
p_j(t) = p_j(0) \frac{p_i(t)}{p_i(0)}
\tag{35}
$$

for all $i, j = 1, 2, \cdots, d$. Then according to (26), we also have $q_j(t) = q_j(0) \frac{p_j(0)}{p_j(t)}$. Substituting them into (34), and we obtain a differential equation with only one variable $p_i$

$$\frac{dp_i}{dt} = -2p_i \left( \frac{(\gamma^2 d)\rho^2 P}{p_i(0)^2} p_i^2 - \frac{(\gamma^2 d)\rho^2 Q p_i(0)^2}{p_i^2} - \gamma \rho y \sqrt{\eta_u \eta_w} \right), \tag{36}$$

where

$$P = \frac{1}{d} \sum_{i=1}^{d} p_i(0)^2, \; Q = \frac{1}{d} \sum_{i=1}^{d} q_i(0)^2. \tag{37}$$

This differential equation is analytically solvable by integration

$$t = -\int_{p_i(0)^2}^{p_i^2} \frac{d\zeta}{4 \left( \frac{(\gamma^2 d)\rho^2 P}{p_i(0)^2} \zeta^2 - \gamma x y \sqrt{\eta_u \eta_w} \zeta - (\gamma^2 d)\rho^2 Q p_i(0)^2 \right)} \tag{38}$$

Because the denominator as a quadratic polynomial has two different roots $\alpha_\pm$, the result of the integration is

$$t = -\frac{t_c}{4} \log \frac{p_i(t)^2/p_i(0)^2 - \alpha_+}{p_i(t)^2/p_i(0)^2 - \alpha_-} + const, \tag{39}$$

leading to

$$\frac{p_i(t)^2/p_i(0)^2 - \alpha_+}{p_i(t)^2/p_i(0)^2 - \alpha_-} = \frac{1 - \alpha_+}{1 - \alpha_-} \exp\left( -4t/t_c \right), \tag{40}$$

which gives (8). $\qquad \square$

**Proposition 2.** *Under the condition in Theorem 1, if $P = 0$ and $Q \neq 0$, the result becomes*

$$p_i(t) = 0 \tag{41}$$

$$q_i(t) = q_i(0) \sqrt{\frac{\alpha' \xi'(t)}{1 - \xi'(t)}} \tag{42}$$

*where*

$$\xi'(t) := \frac{1}{1 + \alpha'} \exp\left( -4\sqrt{\eta_u \eta_w} \gamma \rho y t \right), \tag{43}$$

*and*

$$\alpha' := \frac{\sqrt{\eta_u \eta_w} \gamma \rho y}{(\gamma^2 d)\rho^2 Q}. \tag{44}$$

*Specially, if $P = Q = 0$, we have $p_i(t) = q_{(t)} = 0$, so the gradient flow will be stuck at the trivial saddle point.*

Its proof is similar to the proof of Theorem 1, because we can similarly obtain

$$\frac{dq_i}{dt} = -2q_i \left( \frac{(\gamma^2 d)\rho^2 Q}{q_i(0)^2} q_i^2 + \gamma \rho y \sqrt{\eta_u \eta_w} \right). \tag{45}$$

Its solution gives Proposition 2.

We note that the behavior of the solution is quite different from $P \neq 0$: when $y \leq 0$, we can obtain a solution with zero loss in the end, but when $y > 0$, the gradient flow will converge to the trivial saddle point $p_i = q_i = 0$.

### B.3 PROOF OF THEOREM 2

*Proof.* By definition, $\xi(t)$ is a monotonic function. As $\frac{\alpha_+ - \xi \alpha_-}{1 - \xi} = \frac{\alpha_+ - \alpha_-}{1 - \xi} + \alpha_-$ is monotonous to $\xi$, it evolves from 1 to $\alpha_+$ monotonously. Then according to Equation (13), $\lim_{\kappa \to \infty} K(x, x')(t) = \lim_{\kappa \to \infty} K(x, x')(0)$ if and only if $\lim_{\kappa \to \infty} \alpha_+ = 1$, which holds if and only if $\lim_{\kappa \to \infty} P/Q = 1$ and (19) holds, when we have

$$\lim_{\kappa \to \infty} \alpha_+ = \lim_{\kappa \to \infty} \frac{2\rho^2(\gamma^2 d)\sqrt{PQ}}{2\rho^2(\gamma^2 d)P} = 1. \; a.s. \tag{46}$$

From Equation (18), Equation (19) also implies

$$2c_\gamma + c_d + \max\{c_{\eta_w} + c_u, c_{\eta_u} + c_w\} = 0. \tag{47}$$

Therefore, we can see that the NTK remains $\Theta(1)$ because

$$\gamma^2 d\alpha_+ P = \Theta\big(\kappa^{2c_\gamma + c_d + \max\{c_{\eta_w} + c_u, c_{\eta_u} + c_w\}}\big) = \Theta(1). \tag{48}$$

The proof is complete. $\qquad\square$

### B.4 SCALING OF WEIGHT EVOLUTION

This section aims to quantitatively analyze the power-law scaling of the right side of Figure 4. (7) and (8) indicate that

$$\frac{w_{ij}(+\infty) - w_{ij}(0)}{w_{ij}(0)} = \frac{p_i(+\infty) + q_i(+\infty)}{p_i(0) + q_i(0)} - 1 = \frac{\sqrt{\alpha_+}\, p_i(0) + q_i(0)/\sqrt{\alpha_+}}{p_i(0) + q_i(0)} - 1. \tag{49}$$

In the feature learning regime, $|\alpha_+ - 1| = O(1)$, and thus the weights evolve in an $O(1)$ amount. In the kernel regime, by using (11), (18) and Theorem 2, we can find that

$$|1 - \alpha_+| \propto \kappa^{(2c_\gamma + c_{\eta_u} + c_{\eta_w})/2}. \tag{50}$$

When $c_w + c_u/2 = c_u + c_w/2$, we have $\frac{p_i(0) - q_i(0)}{p_i(0)} = O(1)$ and thus

$$\left| \frac{w_{ij}(+\infty) - w_{ij}(0)}{w_{ij}(0)} \right| \propto |1 - \alpha_+| \propto \kappa^{(2c_\gamma + c_{\eta_u} + c_{\eta_w})/2}. \tag{51}$$

When $c_w + c_u/2 \neq c_u + c_w/2$, we have

$$\left| \frac{w_{ij}(+\infty) - w_{ij}(0)}{w_{ij}(0)} \right| \approx \sqrt{\alpha_+} + 1/\sqrt{\alpha_+} - 2 \approx |1 - \alpha_+|^2 \propto \kappa^{2c_\gamma + c_{\eta_u} + c_{\eta_w}}. \tag{52}$$

In conclusion, we have $\frac{\|W - W_0\|}{\|W_0\|} \propto \kappa^{-\delta}$. In the feature learning regime $\delta = 0$. In the kernel regime $\delta = -\frac{1}{2}(2c_\gamma + c_{\eta_u} + c_{\eta_w})$ if $c_w + c_u/2 = c_u + c_w/2$ and $\delta = -(2c_\gamma + c_{\eta_u} + c_{\eta_w})$ if $c_w + c_u/2 \neq c_u + c_w/2$. Using the values in Table 1, we can verify that $\delta = 0.5, 1, 2$ for NTK, Xavier$^-$ and Kaiming$^-$ parameterization, respectively.

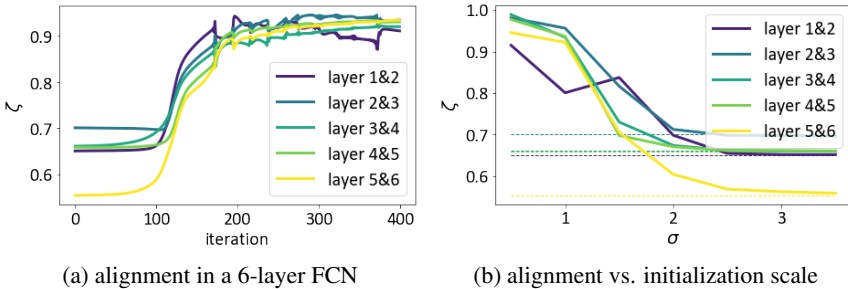

(a) alignment in a 6-layer FCN      (b) alignment vs. initialization scale

Figure 6: The alignment angle $\zeta$ between different layers of a six-layer FCN trained on MNIST, with the same settings as Figure 2.

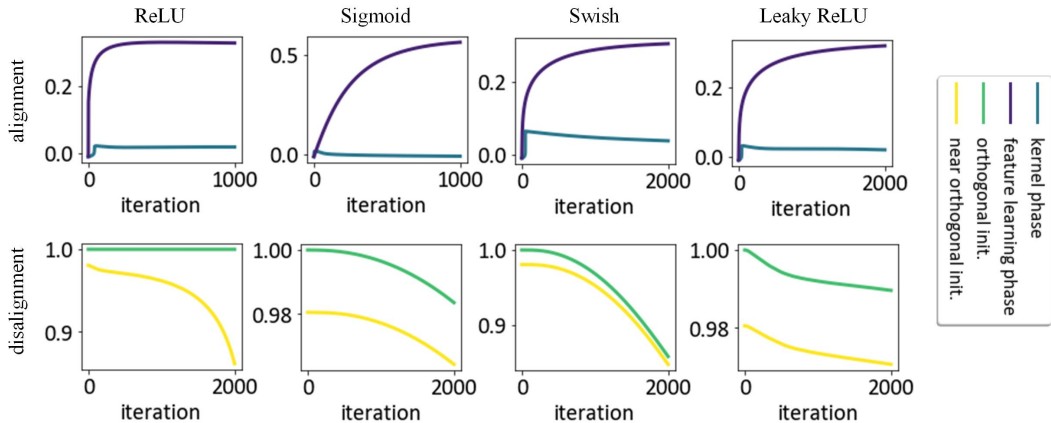

Figure 7: The evolution of the alignment angle $\zeta$ between $u$ and $v$ across two-layer ReLU, sigmoid, swish, and leaky ReLU networks with $d = 10000$. The task is to classify two Gaussian distributions.

## C    ADDITIONAL EXPERIMENTAL CONCERNS

### C.1    ADDITIONAL EXPERIMENTS IN SECTION 3.2

In Figure 1, we choose $x = 1, y = 2$, and for others we randomly sample 100 points from $\mathcal{N}(0,1)$ as data points $x$, and set $y = 2x + \mathcal{N}(0,1)/10$ as the target. The learning rates are chosen such that the model converges well within given iterations. For the orthogonal initialization, we initialize the model as $u \sim \mathcal{N}(0, 10I_d)$ and $w \sim u + \mathcal{N}(0, 0.1I_d)$.

In Figure 2, to avoid the implicit bias of SGD to make layers aligned (Ziyin et al., 2024), we consider full-batch GD with batch size 2000 and constant learning rate. The learning rates are chosen separately for each model such that the model converges well in 1000 iterations, with training accuracy above 95%. All models use the standard Kaiming initialization, but we scale each layer by $\sigma$. The results in Figure 2 also extend to deeper networks, although the training dynamics of deeper FCNs are less stable, as shown in Figure 6.

Moreover, we observe qualitatively the same phenomenon for all kinds of activation functions in the classification task in Figure 7, where the task is to classify training samples from $\mathcal{N}(0,1)$ and $\mathcal{N}(4,1)$. Initialization is the same as in Figure 1, but the binary cross-entropy loss is used. From Figure 7 we can also see that layers tend to align in the feature learning regime when they are initialized to be disaligned, and vice versa. Note that because of the binary cross-entropy loss, $\zeta$ keeps decreasing even after the loss converges. Further, because of the binary cross-entropy loss, $\zeta$ deviates from one for non-linear activation functions other than ReLU.

## C.2 Additional Experiments Concerning the Alignment Effect

Figure 8 verifies that the influence of dataset size $N$ and input dimension $d_0$ on the alignment effect is not significant. This is consistent with theoretical results, because Theorem 1 characterizes the training dynamics without assumptions on the training set size, distribution, or input dimensions.

Figure 9 includes more detailed ablation experiments, including the influence of large learning rate, data not lying in a 1D subspace and a three-layer linear network. Together with other figures (e.g. Figure 2), all ablation experiments indicate that our results are not significantly weakened by the following constraints: 1. 1-hidden layer vs multiple hidden layers, 2. linear vs nonlinear activations, 3. single example (or examples in a 1-dimensional subspace) vs examples distributed through space, 4. gradient flow training vs discrete-time gradient descent or stochastic gradient descent.

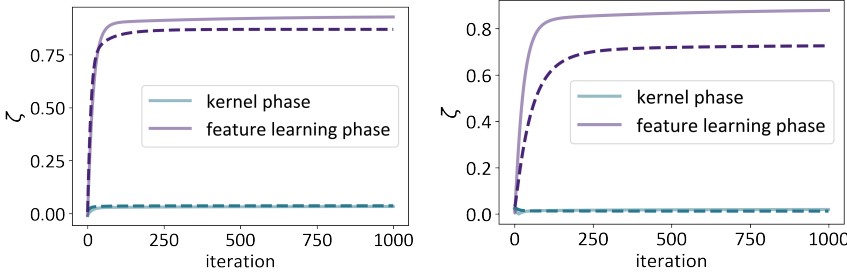

Figure 8: The alignment effect of a ReLU network for different dataset sizes and input dimensions. **Left**: the dashed line represents dataset size 10 and the solid line represents dataset 100. **Right**: the dashed line represents input dimension $d_0 = 10$ and the solid line represents input dimension $d_0 = 1$. The target is chosen to be $y = \alpha^T x + \mathcal{N}(0,1)/10$, where $\alpha$ is a Gaussian vector. Other settings are the same as Figure 1.

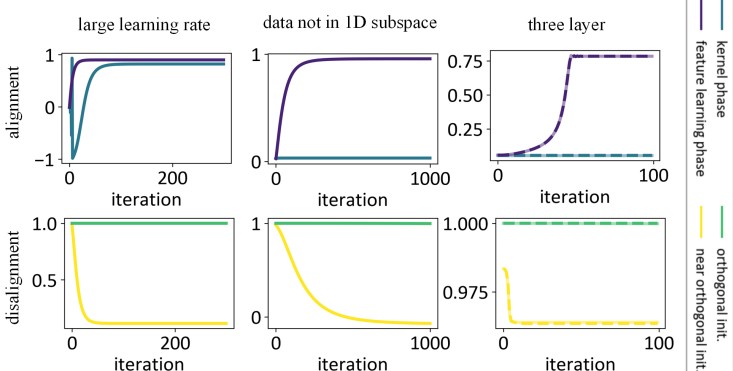

Figure 9: More ablation experiments on linear networks. Results about non-linear activation are similar, as in Figures 1 and 2. **Left**: Learning rate 15 times that of Figure 1. The network jumps out of the kernel regime as predicted by Lewkowycz et al. (2020), but the alignment and disalignment effects still exist. **Middle**: $x$ is a Gaussian vector with $d_0 = 10$, so the data are not in a 1-dimensional subspace. For the orthogonal initialization, the first column of $W$ is the same as $u$ and the others are zero. **Right**: a three-layer linear network with $\gamma = 1/d^2$ and $d = 300$. The solid line refers to the alignment between the first two layers, and the dashed line refers to the alignment between the last two layer. For the orthogonal initialization, the first row of the second layer is the same as the first layer and the others are zero. The last layer is the same as the first column of the second layer. Other settings are the same as Figure 1. This figure, together with Figures 1 and 2, shows that our results are robust to all factors.

## C.3 Experiments in Section 3.3

In Figure 3, we train a Resnet18 network on the CIFAR-10 dataset with hyperparameters borrowed from https://github.com/kuangliu/pytorch-cifar. The only difference is that we scale each layer by $\sigma$ and record the test accuracy together with the sum of the norm of all layers.

## C.4 Experiments in Section 4.1

In Section 4.1, we utilize a two-layer FCN with the ReLU activation and $d$ hidden units. The input is vectorized and normalized, so the input dimension is $d_0 = 3072$. The cross-entropy loss and the

stochastic gradient descent without moment or weight decay are used during training. We use a batch size of 128 and report the best training and test accuracy among all epochs.

We choose $\gamma = \frac{1}{\sqrt{d}}$ and $\eta = 0.05$ for the standard NTK model, $\gamma = \frac{10}{d}$ and learning rate $\eta = 0.05d/100$ for the standard mean-field model, $\gamma = 1$ and $\eta = 0.05d/100$ for the Kaiming$^-$ model, $\gamma = \frac{100}{d}$ and $\eta = 0.05d/100$ for the Kaiming$^+$ model, $\gamma = 1$ and $\eta = 0.05$ for the Xavier$^+$ model, $\gamma = 0.01d$ and $\eta = 0.05(100/d)^2$ for the Xavier$^-$ model. The choice of hyperparameters guarantees that the standard NTK model and the standard mean-field model, the Kaiming$^+$ and Kaiming$^-$ model, and the Xavier$^+$ and Xavier$^-$ model are the same for $d = 100$, respectively.

### C.5   EXPERIMENTS IN SECTION 4.2

The experiment in Section 4.2 is similar to that in 4.1. The only difference is that we fix $d = 2000$ and change the initialization scale. More specifically, we set $\kappa = 10$, $\sigma_u^2 = \kappa^c$, $\sigma_w^2 = \kappa^{\max\{c,0\}}$ and $\gamma = \kappa^{-\min\{0,-c/2\}}$. We also fix $\eta = 0.005$.

## D   TRAINING AND GENERALIZATION DYNAMICS

This section aims to present the evolution of empirical and population loss.

From the definition of $p_i$ and $q_i$ in Theorem 1, we have

$$\frac{\sqrt{\eta_u \eta_w}}{\rho} \sum_{j=1}^{d_0} u_i(t) w_{ij}(t) x_j = p_i(t)^2 - q_i(t)^2. \tag{53}$$

Consequently, we have

$$\gamma \sum_{i=1}^{d} \sum_{j=1}^{d_0} u_i(t) w_{ij}(t) x_j = \frac{\gamma \rho}{\sqrt{\eta_u \eta_w}} \left( P \left[ \frac{\alpha_+ + \xi(t)\alpha_-}{1 - \xi(t)} \right] - Q \left[ \frac{\alpha_+ + \xi(t)\alpha_-}{1 - \xi(t)} \right]^{-1} \right). \tag{54}$$

As $\xi(t)$ is monotonous, $\frac{\alpha_+ + \xi(t)\alpha_-}{1 - \xi(t)}$ monotonously evolves from $1$ to $\alpha_+$. Therefore, the model $\gamma \sum_{i=1}^{d} \sum_{j=1}^{d_0} u_i(t) w_{ij}(t) x_j$ monotonously evolves from the initial value $\frac{\gamma \rho d}{\sqrt{\eta_u \eta_w}}(P - Q)$ to the final value

$$\frac{\gamma \rho d}{\sqrt{\eta_u \eta_w}}(\alpha_+ P - Q/\alpha_+) = \frac{\gamma \rho d}{\sqrt{\eta_u \eta_w}}(\alpha_+ + \alpha_-)P = y, \tag{55}$$

where we use the definition of $\alpha_+, \alpha_-$ and $\alpha_+ \alpha_- = -Q/P$.

In conclusion, the empirical loss (see Appendix A.1)

$$\tilde{L}(u, w) = [\gamma \sum_{i=1}^{d} \sum_{j=1}^{d_0} u_i(t) w_{ij}(t) x_j - y]^2 + \sum_{k=1}^{N} y_k^2 - y^2 \tag{56}$$

evolves from its initial value to its minimal value $\sum_{k=1}^{N} y_k^2 - y^2$ monotonously. Notably, the above conclusion does not rely on the choice of all hyperparameters and initialization.

In terms of the population loss, we note that when the data $x := an$ lies in a one-dimensional subspace, the model output can be written as a linear function $f(an) =: as(t)$, where $s$ is a scalar parameter. The population loss is thus a quardratic function of $s$: $\mathbb{E}[(as - \tilde{y})^2]$, which takes the minimal at

$$\frac{\mathbb{E}[a\tilde{y}]}{\mathbb{E}[a^2]}.$$

In reality, however, $s(t)$ evolves monotonously from its initial value to

$$\frac{\sum_{k=1}^{N} a_k \tilde{y}_k}{\sum_{k=1}^{N} a_k^2}.$$

According to the initial value of $t$ and the dataset $a_k, \tilde{y}_k$, there are three cases. The population loss might monotonously decrease, monotonously increase, or first decrease and then increase. The time

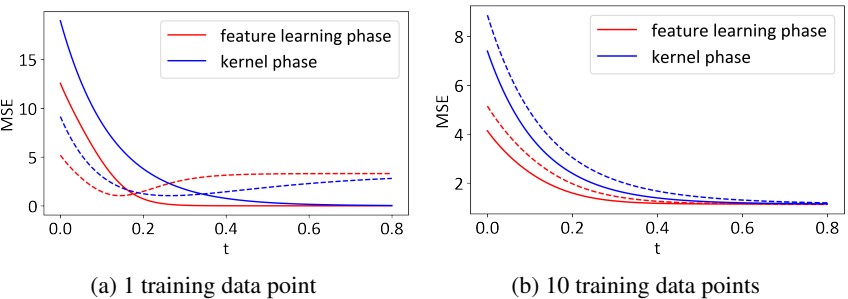

(a) 1 training data point            (b) 10 training data points

Figure 10: The evolution of the training loss (solid lines) and the generalization error (dashed lines) for a linear network. We choose $x = 1, y = 2 + \mathcal{N}(0, 1)$.

scale of the population loss is also $t_c$, so there is no grokking for this simple model. Finally, as $N$ increases, $\frac{\sum_{k=1}^N a_k \tilde{y}_k}{\sum_{k=1}^N a_k^2} \to \frac{\mathbb{E}[a\tilde{y}]}{\mathbb{E}[a^2]}$, and the population loss converges to its minimum as expected.

See Figure 10 for numerical verification. The results include two possibilities of the generalization error: monotonously decrease, or first decrease and then increase, as predicted theoretically. The time scales of the training loss and the generalization error are also the same. Moreover, we can see that for different initialization and parameterization, the networks converge to the same MSE. Therefore, our main focus is the training dynamics (e.g., whether NTK evolves), and our main contribution lies in analyzing how these dynamics unfold rather than in the final convergence point. The results also indicate that the number of samples has no significant impact on the training dynamics but will influence the generalization dynamics.

