# OpenReview forum: "Three Mechanisms of Feature Learning in a Linear Network"
_ICLR.cc/2025/Conference — ICLR 2025 Poster_

### Official Review · Reviewer_T1qw · 2024-10-30

**Soundness:** 3
**Presentation:** 2
**Contribution:** 2
**Rating:** 6
**Confidence:** 4

**Summary:**

This work proposes a solvable shallow (linear) network to gain insight into feature/kernel-regime classification and its parameterization. For a one-dimensional data space, the continuous-time learning dynamics can be reduced to certain variables $(p_i, i=1, ..., \text{width})$, allowing for explicit solutions in a specific case $(\beta=1)$. From this solution, particularly from the learning speed $(t_c^{-2})$, the authors identify the parameterization in as general a form as possible, in which the learning dynamics achieve either the feature-learning or kernel regime.

**Strengths:**

- The analysis is technically solid and concrete.
- This work addresses the challenging yet crucial problem of identifying the feature-learning and kernel regimes and characterizing the dynamics in feature learning. It provides concrete insights from a solvable model perspective.
- The obtained parameterizations are quite general which cover various width, learning rate, and initialization scales.

**Weaknesses:**

- The analyzed model seems essentially equivalent to the catapult model (Lewkowycz et al. 2020) or uv model (Kalra et al. 2023). In this sense, the solvability of this model itself seems not so surprising. In addition, It is unclear how the continuous-time analysis of the current work contributes to the original literature of the catapult (uv) model, which claims that a non-trivial dyanamics toword feature learning appears for *discrete* update with a large learning rate.

- Certainly, discrete updates have not been explicitly solved and achieving analytical solvability even in the continuous limit would be novel. However, for alignment, the dynamics is essentially equal to the uv product dynamics, which can already be tracked with any width (Lewkowycz et al. 2020). Therefore, the analytical solvability for the finite width might be a bit straightforward.

- Although this would be a presentation matter, it is not easy to follow how the parameterization derived from the current analysis in Section 4.1 compares to the muP parameterization by Yang & Hu (2020).

Please see the **Questions** section for more detailed points.

**Questions:**

### **Difference from uv model**
How does the proposed model differ from the well-known catapult (uv) model? Suppose that we rotate the 1-dim data space $n$ to the direction of a unit vector $(1,0,\dots,0)$ with a rotation matrix $V$. Then, defining the new variable $W(t)= W^*(t) V$, we have:
$\frac{dW}{dt} = -\frac{\partial L}{\partial W}$
equivalent to
$\frac{dW^*}{dt} = -\frac{\partial L}{\partial W^*}$
since the rotational transformation is orthogonal (that is, the Jacobian $J=\partial W^*/\partial W$ satisfies $J^T J=I$). In other words, the proposed model in this work reduces to

$L(u, W^*) = \left[\gamma \left(\sum_{i} u_i w^{*\beta}_i  x  - y \right)\right]^2$

with $x:=\sqrt{E[\tilde{x}^2]}^\beta$.

This suggests that the proposed model is merely a nonlinear parameterization of the uv model, with $v_i = w_i^\beta$. For $\beta=1$, which is the focus of the current work, the model is identical to the uv model. Since the authors references Lewkowycz et al. (2020), they  likely know about the solvability of the uv model, yet there is no explanation of this relationship or reference to further developments such as

*Kalra et al., "Universal Sharpness Dynamics in Neural Network Training: Fixed Point Analysis, Edge of Stability, and Route to Chaos", arXiv:2311.02076.

*Kalra & Barkeshli, "Phase diagram of early training dynamics in deep networks: effect of the learning rate, depth, and width", NeurIPS 2023.

Since  the model is indeed equivalent to the uv model for $\beta=1$, it is crucial to mention such prior studies and to carefully explain in what aspects the problem setting differs.

### **Difference from muP**
- It will be better to provide a clearer explanation in Corollaries 1-3 about how these findings differ from the muP parameterization [Yang & Hu (2020)]. The authors state
>one can choose an optimal pair of learning rate and output scale $\gamma$ such that the model is in the feature learning phase. Point (1) agrees with the analysis in Yang & Hu (2020), whereas point (2) is a new insight we offer

 in lines 471-473. My understanding is that point (2) would be a main difference (in the inifinite width). However, $\gamma$ also appears in muP as $1/(width)^a$ in the final layer. I recommend that the authors more explicitly state how their "optimal pair of learning rate and output scale γ" differs from or extends the muP parameterization.

If the claim of this work is that the authors have identified a previously unknown feature-learning (or kernel) regime in the infinite-width case, this would imply that certain conditions in prior studies were restrictive, leading to oversight. It seems better to clearly explain which specific conditions from prior work the current work has relaxed or generalized.

- The authors also state
>A key difference mentioned is that the results apply to finite-width networks with arbitrary initialization.

This sentence mentioning the finite width appears in Section 4.1, which consider the case of $c_d=1$, that is, the *infinite* width limit. This is confusing. If there's a distinction between their finite-width and infinite-width results, clearly delineate which parts of the paper apply to each case.

### **Typo**
- Equation 1: $x \to \tilde{x}$

###After the rebuttal########

I raised the score because the author(s) revised unclear points and also promised to explain related research in essentially the same model (but different focus) in the final version.

---

> ### Author Response · Authors · 2024-11-16
>
> # Response to Comment 1
>
> * *The analyzed model seems essentially equivalent to the catapult model (Lewkowycz et al. 2020) or uv model (Kalra et al. 2023). In this sense, the solvability of this model itself seems not so surprising. In addition, It is unclear how the continuous-time analysis of the current work contributes to the original literature of the catapult (uv) model, which claims that a non-trivial dyanamics toword feature learning appears for discrete update with a large learning rate. Certainly, discrete updates have not been explicitly solved and achieving analytical solvability even in the continuous limit would be novel. However, for alignment, the dynamics is essentially equal to the uv product dynamics, which can already be tracked with any width (Lewkowycz et al. 2020). Therefore, the analytical solvability for the finite width might be a bit straightforward.*
>
> Thank you for your insightful feedback. The references you mentioned are very instructive and we  have added them to the related literature section. While our model shares some similarities with the catapult model (Lewkowycz et al., 2020), there are three critical differences that set our work apart
>
> * **Scaling Regime**: The catapult model adopts NTK parameterization ($c_\gamma = -1/2$), placing it in the kernel regime when considering gradient flow or sufficiently small learning rates. In contrast, our work analyzes a setting where feature learning mechanisms are present even in the gradient flow limit. Thus, the feature learning dynamics we capture do not appear in the catapult model due to its NTK scaling.
>
> * **Mechanism of Exiting the Kernel Regime**: Lewkowycz et al. (2020) and subsequent works propose that a neural network can exit the kernel regime with a sufficiently large learning rate. In stark contrast, our analysis shows that changing the parameterization, even while keeping the learning rate small, can also drive the model out of the kernel regime. These are two fundamentally distinct mechanisms, highlighting a novel perspective in our work.
>
> * **Analytical Solvability**: The catapult (uv) model does not provide an explicit analytical solution for the learning dynamics. Lewkowycz et al. (2020) rely on asymptotic expansions at O(width^{-1}), making their results approximate and valid only in the large-width limit. Similarly, while Kalra et al. (2023) extend the analysis to finite-width settings with different parameterizations, their results remain qualitative, perturbative, or focus solely on fixed points without deriving a complete analytic solution. In contrast, our work provides a fully solvable model with exact dynamics, even for finite widths, under more general initialization and parameterization settings.

---

> ### Author Response · Authors · 2024-11-16
>
> # Response to Comment 2
>
> * *It will be better to provide a clearer explanation in Corollaries 1-3 about how these findings differ from the muP parameterization [Yang & Hu (2020)]. The authors state that “one can choose an optimal pair of learning rate and output scale such that the model is in the feature learning phase. Point (1) agrees with the analysis in Yang & Hu (2020), whereas point (2) is a new insight we offer” in lines 471-473. My understanding is that point (2) would be a main difference (in the inifinite width). However, \gamma also appears in muP as 1/(width)^a in the final layer. I recommend that the authors more explicitly state how their "optimal pair of learning rate and output scale γ" differs from or extends the muP parameterization.*
>
> * *If the claim of this work is that the authors have identified a previously unknown feature-learning (or kernel) regime in the infinite-width case, this would imply that certain conditions in prior studies were restrictive, leading to oversight. It seems better to clearly explain which specific conditions from prior work the current work has relaxed or generalized.*
>
> * *The authors also state “a key difference mentioned is that the results apply to finite-width networks with arbitrary initialization.” This sentence mentioning the finite width appears in Section 4.1, which consider the case of c_d=1, that is, the infinite width limit. This is confusing. If there's a distinction between their finite-width and infinite-width results, clearly delineate which parts of the paper apply to each case.*
>
> We appreciate your comments. We believe that it is due to a misunderstanding concerning Corollaries 1-3. To address your points, we have moved Corollaries 1-3 out of Section 4.1 to highlight that **they are not restricted to the infinite-width setting**. These corollaries are applicable to **both infinite-width ($c_d = 1$) and finite-width ($c_d = 0$) cases**, as demonstrated in Sections 4.1 and 4.2, respectively. This is why we state that our results apply to finite-width networks with arbitrary initialization, and are thus an extension to muP parameterization. By positioning the corollaries more broadly in the revised manuscript, we aim to clarify their general applicability.
>
> # Response to Comment 3
>
> * *Typo. Equation 1: x\to\tilde{x}.*
>
> Thank you for pointing this out. We have corrected this typo.

---

> > ### Author Response · Authors · 2024-11-25
> >
> > Hi! We noticed that you have not communicated with us during the discussion period. It would be really helpful to us when revising and improving our manuscript if we could hear more from you. We would be happy to hear any additional thoughts or feedback you have on our work and on our previous reply!

---

> ### Comment · Reviewer_T1qw · 2024-11-27
> **Thank you for your clarification**
>
> The author's revised manuscript has addressed my concerns, so I will increase my score.
>
> That said, I strongly recommend explicitly adding the equivalence to the catapult (uv) model I mentioned. Specifically, I suggest explicitly writing the equivalence by showing the coordinate transformation in the final manuscript. Currently, the authors just describe "some similarity with our model, their focus is completely different."
> However, this explanation feels indirect, and I suspect general readers cannot understand the precise meaning of similarity. While I understand that *the focus* is different, highlighting the equivalence to *the model* would be highly informative for subsequent studies to better connect with your work.
>
> Additionally, it may be better to discuss the following point in the final section or the appendix:
> While I understand that the current contribution primarily emphasizes the finite-width dynamics,  previous studies on the catapult effect and similar phenomena emphasize that discrete updates with a large learning rate have a bias toward feature learning (which has been also empirically verified well). It will be beneficial to highlight this distinction explicitly.

---

> ### Author Response · Authors · 2024-11-28
>
> Thank you for increasing the score! We will surely incorporate these changes in the final manuscript, especially emphasizing the equivalence to the uv model and the fact that discrete updates with a large learning rate bias toward feature learning, to help readers better appreciate the connections and distinctions between our work and prior studies.

---

### Official Review · Reviewer_9ybf · 2024-11-03

**Soundness:** 3
**Presentation:** 2
**Contribution:** 3
**Rating:** 6
**Confidence:** 3

**Summary:**

The paper studies three mechanisms of feature learning (alignment, disalignment, rescaling) in a one hidden layer (linear) neural network. Under a data assumption, the authors characterize the solution of the network fully in terms of width, initialization scale and learning rate across different regimes. The authors also confirm their findings with empirical evidence.

**Strengths:**

The main strength of the paper lies in Theorem 1, which completely characterizes the dynamics of the layer weights across time, under the assumption that the data lies on a 1D subspace. The authors obtain solutions for general power activations, but focus on giving closed form solutions for the case of a linear network $\beta=1$ due to the problem of computing $F_i^{-1}$. By studying the alignment between the layer weights (in the case of $x$ being 1D), the authors characterize the 3 mechanisms of feature learning - which are governed by the angle $\zeta$. Finally, the authors use the theoretical framework developed to study scaling limits in Section 4.

In general I think the paper is quite thorough and does a great job covering the mechanisms of feature learning in this toy setting. To my knowledge, the theoretical analysis of these dynamics in the finite width case is novel and interesting, bringing a great addition to the community.

**Weaknesses:**

In my opinion the main weakness of this paper is the exposition. There are quite a bit of variables involved in the main body which makes it hard to track which quantity is which. Given that the whole dynamics description in the paper revolves around Theorem 1, it might be worthwhile to restate it in simpler terms.

Another (minor in my opinion, given the analytical intractability of network that is more complicated) weakness is the very simplified setting studied, in particular the restriction over the data lying in a 1D subspace.

**Questions:**

If I’m not mistaken, at line 891, if we take $\beta=1$, and plug it into equation (30), then $F_i(x) = ln(x) + c$, not $e^x$. I believe this should be the inverse of $F_i$? In which case I do not understand how (33) reduces to (35).

Shouldn’t there be $x_{ij}$ with a tilde in Proposition 1? And similarly the loss rewriting in eqn. 3 shouldn’t there be an expectation?
I don’t understand what the authors mean in line 218 i.e. “the dynamics of GD training only has a rank-1 effect”.

If I am not misunderstanding, in Line 209, I believe now there is a vector $w$ instead of the previous matrix $W$ because $x$ is taken to be 1d, but this is only mentioned later on in line 217-218, which is confusing.

Could the authors elaborate what they mean in line 239, more specifically why would a larger norm (assuming of the weights?) imply strong overfitting?

Would it be possible as future work to analyze, in the same framework, the solutions for arbitrary nonlinearities? Maybe by using Hermite polynomials?

Is there any intuition for why the alignment increases much faster for Sigmoid/Swish/LeakyReLU in Figure 1?

---

> ### Author Response · Authors · 2024-11-16
>
> # Response to Comment 1
>
> * *In my opinion the main weakness of this paper is the exposition. There are quite a bit of variables involved in the main body which makes it hard to track which quantity is which. Given that the whole dynamics description in the paper revolves around Theorem 1, it might be worthwhile to restate it in simpler terms.*
>
> Thank you for your valuable feedback. We acknowledge that the complexity of the final solution can make it challenging to follow the variables and their roles. Due to the intricate nature of our results, restating Theorem 1 in a significantly simpler manner is challenging. However, to improve clarity, we have added a summary of all key variables in Table 2 and emphasized the most critical results immediately following Theorem 1 in the revised manuscript. We hope these additions will make the exposition clearer.
>
> # Response to Comment 2
>
> * *Another (minor in my opinion, given the analytical intractability of network that is more complicated) weakness is the very simplified setting studied, in particular the restriction over the data lying in a 1D subspace.*
>
> Thank you for your suggestion. We are fully aware of this weakness, so we have conducted extensive experiments on non-linear networks and real-world datasets to demonstrate that the insights gained from our toy model can extend to more diverse and realistic settings. We have emphasized this in the main text to clarify the broader applicability of our findings.
>
> # Response to Comment 3
>
> * *If I’m not mistaken, at line 891, if we take \beta=1, and plug it into equation (30), then $F_i(x)=\ln x+c$, not $e^x$. I believe this should be the inverse of $F_i(x)$? In which case I do not understand how (33) reduces to (35).*
>
> Thank you for pointing this out. This is indeed a typo. It should be $F_i(x)=\ln x$ and its inverse is $F_i^{-1}(x)=e^x$ (the constant is not important). Therefore, we have $\Delta_i(t)=\ln p_i(t)$. Taking this into (33), and we will obtain (35).
>
> # Response to Comment 4
>
> * *Shouldn’t there be $x_{ij}$ with a tilde in Proposition 1? And similarly the loss rewriting in eqn. 3 shouldn’t there be an expectation? I don’t understand what the authors mean in line 218 i.e. “the dynamics of GD training only has a rank-1 effect.*
>
> Thank you very much for your question. In Proposition 1, $\tilde{x}$ is a random vector while $x:=\sqrt{E[\tilde{x}^2]}$ is a fixed vector. Under the assumption that  $\tilde{x}$ lies in a one-dimensional subspace, we can explicitly do the expectation with respect to $\tilde{x}$, leading to the loss of x eq.(3) without any randomness. See the proof of Proposition 1 for details.
>
> The rank-1 effect refers to the following. When the data lie in a one-dimensional subspace, the assumption that x is 1D does not influence the generality because, in Theorem 1, p and q are two effective weight vectors, which we call the rank-1 effect.

---

> ### Author Response · Authors · 2024-11-16
>
> # Response to Comment 5
>
> * *If I am not misunderstanding, in Line 209, I believe now there is a vector W instead of the previous matrix because x is taken to be 1d, but this is only mentioned later on in line 217-218, which is confusing.*
>
> Thank you for your question. Now we reorganize this part, saying at the beginning that x is taken to be 1d, while numerical results for general input are provided at the end.
>
> # Response to Comment 6
>
> * *Could the authors elaborate what they mean in line 239, more specifically why would a larger norm (assuming of the weights?) imply strong overfitting?*
>
> Norm-based generalization bounds suggest that larger norm leads to worse generalization. See arXiv:1905.12430 for example. An intuition is that the norm of the weights might be less directly related to learning specific features. For example, rescaling will not influence the performance of a ReLU network in classification.
>
> # Response to Comment 7
>
> * *Would it be possible as future work to analyze, in the same framework, the solutions for arbitrary nonlinearities? Maybe by using Hermite polynomials?*
>
> Thank you for your question. I believe that it is possible to analyze as long as the data lie in a 1D subspace, because the gradient flow reduces to a one-dimensional ODE for monomial activation, and probably we can generalize such results to polynomial actications.
>
> # Response to Comment 8
>
> * *Is there any intuition for why the alignment increases much faster for Sigmoid/Swish/LeakyReLU in Figure 1?*
>
> Thank you for your question. The observed differences in alignment rates in Figure 1 were due to using different learning rates for the various activations. In the revision, we have replotted all figures using a consistent learning rate across all activation functions. With this adjustment, we observe that the alignment for linear, Swish, LeakyReLU, and ReLU activations increases at a similar rate, while the alignment for sigmoid increases much slower. This slower rate for sigmoid is expected, as its smaller derivative leads to reduced gradient flow during training.

---

> > ### Comment · Reviewer_9ybf · 2024-11-24
> > **Reply**
> >
> > I thank the authors for their reply and for including my recommendations in their manuscript. I will keep my score.

---

### Official Review · Reviewer_jPRE · 2024-11-03

**Soundness:** 3
**Presentation:** 2
**Contribution:** 3
**Rating:** 6
**Confidence:** 2

**Summary:**

The paper studies a linear network setup with single-index inputs and derives exact learning dynamics under gradient flow, and interprets the resulting solutions. In doing so, the paper finds that the time to learn the target function depends on a kernel and feature learning component and that the feature learning component in particular can come from several mechanisms, which are each interpreted and discussed. This results in insights about the role of unbalanced initializations, and output rescaling, the latter being a quantity central to understanding feature learning from past work.

**Strengths:**

Overall, I liked this paper. The final rating would be even higher if I was confident in my expertise in this topic (linear networks/optimization trajectories). In the end, I am not such an expert, so my rating reflects the fact that I cannot be sure other related works do not present very similar results. But in my opinion, the strong point of this paper was providing 1) an exact set of solutions, in contrast to the usual perturbative approach from infinite-width dynamics (which is usually extremely complicated and does not yield much insight) and 2) spending several pages interpreting the equations that result -- this is very important: the notion that the feature learning component can come from three distinct "types of feature learning" was new and interesting to me.

**Weaknesses:**

- You never write down the dynamics for the NTK in Equation (13), even though as I understand it you claim the theory implies NTK evolution. Maybe this is immediate from your solution on the weight dynamics, but some interpretation for what the evolving kernel means would be useful.
- The word "disalignment" was confusing to me. As I understand it, it just refers to learning due to asymmetry between either the initialization or learning rate for the two layers, where feature learning doesn't take place under symmetry/orthogonal initialization. I'm not sure what "disalignment" means in this context, whereas for instance the meaning of "alignment" is intuitively clear.
- The idea that at some intermediate initialization scale we are not quite in the kernel regime but also not quite aligning layers to the task in some intuitive sense was interesting, and I wish there was more discussion on this topic (ie. Section 3.4).
- The point made in passing at the end about "inadequacy of conservation laws" and the impossibility of stronger analytical results is provocative and insufficiently explained. If you are making such a comment, you must explain it in more detail.
- Minor spelling/grammar, eg. Theorem 1.1 statement "Dynamics" typo
- I think a set of bullet points of contributions in simple (nontechnical) prose would be useful in the Intro or Section 3. There are many claims made throughout and results derived, and it's sometimes not clear what the overall aim of the paper is. Having one bullet point for each section to illustrate 1) what exactly is the new result and 2) how it all relates to each other would be very useful.

**Questions:**

- Am I correct in understanding that $p_i, q_i$ are something like "order parameters" in the parlance of some statistical physics analyses of network dynamics (DMFT, etc)? ie. you are studying the dynamics of some "sufficient statistics" which are informally some rotated/transformed version of our true weights?
- Why don't either $p_i, q_i$ or initialization variables show up in the feature learning component -- what is the intuition for this? Why is there dependence on the input norm $\rho$ here?

---

> ### Author Response · Authors · 2024-11-16
>
> # Response to Comment 1
>
> * *You never write down the dynamics for the NTK in Equation (13), even though as I understand it you claim the theory implies NTK evolution. Maybe this is immediate from your solution on the weight dynamics, but some interpretation for what the evolving kernel means would be useful.*
>
> Thank you for the suggestion. The dynamics of the NTK can indeed be derived by substituting u(t) and w(t) from equations (7)-(11) into equation (13), which we have now included in the manuscript. However, the resulting expressions are quite complex, so we opted not to present them explicitly. Instead, we focus on whether the NTK evolves by an O(1) amount in subsequent sections, such as in Definition 1.
>
> # Response to Comment 2
>
> * *The word "disalignment" was confusing to me. As I understand it, it just refers to learning due to asymmetry between either the initialization or learning rate for the two layers, where feature learning doesn't take place under symmetry/orthogonal initialization. I'm not sure what "disalignment" means in this context, whereas for instance the meaning of "alignment" is intuitively clear.*
>
> Thank you for your question. The term "alignment" in our paper refers to the cosine similarity between two consecutive layers, as defined at the beginning of Section 3.2 for vectors and at the end of Section 3.2 for matrices. This concept is also discussed in Everett et al. (2024). In this context, “alignment” implies that consecutive layers are more parallel, while “disalignment” indicates they are closer to being orthogonal.
>
> Intuitively, alignment is preferable because aligned layers require a smaller model norm to achieve the same prediction accuracy, while disaligned layers necessitate a much larger model norm. This is consistent with our observation in the kernel phase, where alignment remains zero (Figure 1). Additionally, as noted in the discussion following equation (16), disalignment is indeed more likely when the layers exhibit greater asymmetry, such as in initialization or learning rates.
>
> # Response to Comment 3
>
> * *The idea that at some intermediate initialization scale we are not quite in the kernel regime but also not quite aligning layers to the task in some intuitive sense was interesting, and I wish there was more discussion on this topic (ie. Section 3.4).*
>
> Thank you for your suggestion. The nuanced behavior you mentioned arises because, within the feature learning phase, the model can exhibit both alignment and disalignment, suggesting that there are perhaps multiple modes of feature learning (e.g., alignment-dominated vs. disalignment-dominated). When the initialization is relatively small, the model is in the feature learning regime, with learning driven primarily by alignment. At intermediate initialization scales—large, but not excessively so—the model still operates in the feature learning regime but begins to learn by disalignment, which can reduce performance. As the initialization scale becomes very large, the model transitions to the kernel regime, where neither alignment nor disalignment plays a role (i.e., \zeta(t) remains constant, as shown in Figure 1). A more detailed, systematic analysis of these different regimes is an intriguing direction for future research.
>
> # Response to Comment 4
>
> * *The point made in passing at the end about "inadequacy of conservation laws" and the impossibility of stronger analytical results is provocative and insufficiently explained. If you are making such a comment, you must explain it in more detail.*
>
> Thank you for highlighting this point. Our comment on the “inadequacy of conservation laws” refers to the result by Marcotte et al. (2023, Corollary 4.4), which suggests that the maximum number of independent conservation laws can be significantly fewer than the degrees of freedom in the system. This implies that, even with conservation laws, solving the high-dimensional ODE analytically remains extremely challenging. We have expanded on this explanation in the manuscript to provide more clarity.

---

> ### Author Response · Authors · 2024-11-16
>
> # Response to Comment 5
>
> * *Minor spelling/grammar, eg. Theorem 1.1 statement "Dynamics" typo*
>
> Thank you for pointing this out. We have corrected this typo.
>
> # Response to Comment 6
>
> * *I think a set of bullet points of contributions in simple (nontechnical) prose would be useful in the Intro or Section 3. There are many claims made throughout and results derived, and it's sometimes not clear what the overall aim of the paper is. Having one bullet point for each section to illustrate 1) what exactly is the new result and 2) how it all relates to each other would be very useful.*
>
> Thank you for your suggestion. To make the contributions clearer, we have now organized them as follows:
>
> * We analytically solve the evolution dynamics of the NTK for a minimal finite-width model with arbitrary initialization. The model we analyze is a one-hidden-layer linear network, which, despite its simplicity, has a non-convex loss landscape and strongly coupled dynamics. Prior to our work, the exact solution for its learning dynamics was unknown.
> * Based on our exact solutions, we identify three novel mechanisms of learning that are exclusive to the feature learning phase of the network.
> * Using our exact solutions, we provide phase diagrams that distinguish between the kernel phase and the feature learning phase.
> * We empirically validate the three mechanisms of feature learning and our phase diagrams in realistic nonlinear networks.
>
> # Response to Comment 7
>
> * *Am I correct in understanding that pi,qi are something like "order parameters" in the parlance of some statistical physics analyses of network dynamics (DMFT, etc)? ie. you are studying the dynamics of some "sufficient statistics" which are informally some rotated/transformed version of our true weights?*
>
> Thank you for your insightful question. We agree that {q_i} look like order parameters because they represent the degree of rescaling symmetry breaking, but perhaps it is better to interpret {p_i,q_i} as generalized coordinates that describe the system at the granular level.
> Other order parameters like $\zeta(t)$ (analogous to overlaps in statistical physics) and the NTK are also important in our analysis because they encapsulate the macroscopic behavior of the system. We analyze them respectively in Sections 3 and 4. Specifically, the NTK can be seen as a form of sufficient statistics since it fully characterizes the evolution of the function f(x), as discussed in the line preceding equation (13).
>
> # Response to Comment 8
>
> * *Why don't either pi,qi or initialization variables show up in the feature learning component -- what is the intuition for this? Why is there dependence on the input norm here?*
>
> The reason that the specific initialization variables do not explicitly appear in the feature learning component is that their effects are fully captured by the sufficient statistics P, Q, and $\sum p_i(0) q_i(0)$. This is demonstrated in eqs (8)-(10) and (14). These sufficient statistics summarize the influence of the initial conditions on the system's dynamics, effectively reducing the dimensionality of the problem in terms of relevant variables.
>
> Regarding the dependence on the input norm, this arises naturally because increasing the input norm is equivalent to increasing the scaling factor $\gamma$. This is why the terms $\gamma\rho$ appears together in our solutions eqs.(10) and (11).

---

> > ### Comment · Reviewer_jPRE · 2024-11-23
> > **Response to Authors**
> >
> > I thank the authors for their response and am writing this to acknowledge it. I will keep my score.

---

### Official Review · Reviewer_Wsc1 · 2024-11-03

**Soundness:** 4
**Presentation:** 3
**Contribution:** 2
**Rating:** 8
**Confidence:** 4

**Summary:**

This paper studies the role of weight initialization on feature learning dynamics in linear networks. The authors provide exact solutions for  two layer linear networks where the input data is one dimensional. In this case, the relevant initial condition is the alignment $\zeta$ which measures the cosine similarity between the readin and readout vectors. Depending on the scale of the initial weights, the scale of the output multiplier, and the initial alignment between these vectors, the model may or may not exit the kernel regime when fitting the data. The authors show that large weights can lead to worse alignment compared to starting with small weights. They provide a list of possible weight, learning rate and output multiplier scalings that can lead to kernel or feature learning.

**Strengths:**

This paper studies an important problem of how model initialization impacts learning dynamics in neural networks. They provide exact solutions for the training in a specific setting (linear networks on one dimensional inputs). The solutions indicate a crucial dependence on output scale, learning rates, initialization variance, etc. The authors connect their findings to kernel limits, mean field limits, and $\mu$P scalings.

**Weaknesses:**

The assumptions of the model make the scope of the theoretical results fairly limited since it relies on one-dimensional inputs and two layer linear networks (though the authors also have some results for other activations). That said, the authors do make an effort to test their ideas in more realistic settings, which is appreciated. There are also some questions which should be addressed (see below).

**Questions:**

1. The authors claim larger initialization leads to worse performance. They then provide an experiment with residual skip connections (Figure 3) to try verifying this claim. I agree that with residual skip connections, networks can be initialized with zero weight variance with little consequence to training speed or performance. However, a non-residual feedforward network (such as the linear networks the authors actually analyze) the network is stuck at a saddle point as the weight variance shrinks to zero, leading to slower optimization. This is the reason for the long escape times in Saxe et al 2013.
2. Are the results/scalings of Table 1 covered by the $a,b,c$ symmetries in the Yang et al [paper](https://arxiv.org/abs/2011.14522) or in this [note](https://mlschool.princeton.edu/sites/g/files/toruqf5946/files/documents/Princeton___Lecture_Notes_0.pdf)? If so, it would be good to comment on.
3. The authors claim that they characterize the role of width. How does width enter their formulas, is it primarily through the initial alignment $\zeta$? Or is width to be interpreted as a possible $\kappa$ scaling quantity?
4. Could Figure 4 be plotted on a log-log scale? I suspect that $|W-W_0| / |W_0 |$ should scale as a power law in $d$ with exponent that is different for different parameterizations.
5. In Figure 5, it appears that the best performance is not at *very small* weights, but rather there is some optimal intermediate weight scale. How do the authors reconcile this
6. Generally throughout the paper some measured metrics are implicitly related to performance/generalization, without this connection being made clear. For example "In the kernel phase, the model norm diverges and the model alignment is always zero, which could be a hint of strong overfitting."

---

> ### Author Response · Authors · 2024-11-16
>
> # Response to Comment 1
>
> * *The authors claim larger initialization leads to worse performance. They then provide an experiment with residual skip connections (Figure 3) to try verifying this claim. I agree that with residual skip connections, networks can be initialized with zero weight variance with little consequence to training speed or performance. However, a non-residual feedforward network (such as the linear networks the authors actually analyze) the network is stuck at a saddle point as the weight variance shrinks to zero, leading to slower optimization. This is the reason for the long escape times in Saxe et al 2013.*
>
> Thank you for your insightful comment. You are correct that initializing weights too close to zero can lead to long training times, especially in non-residual feedforward networks due to saddle points. However, our point is not that the initialization should be as small as possible, but rather that overly large initializations lead to learning via disalignment, whereas smaller (but not infinitesimal) initializations promote alignment. For instance, the parameterization in Yang et al. (2022) ensures O(1) activations at initialization, which should facilitate alignment in learning. This aligns with our recommendation of using small—but not excessively small—initializations. We have clarified this point in the revision.
>
> # Response to Comment 2
>
> * *Are the results/scalings of Table 1 covered by the abc symmetries in the Yang et al paper or in this note? If so, it would be good to comment on.*
>
> Thank you for pointing this out. Table 1 is consistent with the abc symmetries in Yang et al. (2022). As noted in our revision, equations (9)-(11) demonstrate that the solution remains invariant under the transformation: $u \rightarrow Cu, w \rightarrow Cw, \gamma \rightarrow C^{-2} \gamma, \eta_u \rightarrow C^2 \eta_u​, \eta_w \rightarrow C^2 \eta_w​$, leading to $P \rightarrow C^4 P$ and $Q \rightarrow C^4 Q$. This corresponds to the scaling transformation $c_u \rightarrow c_u + \theta, c_w \rightarrow c_w + \theta, c_\gamma \rightarrow c_\gamma - 2\theta, c_{\eta_u} \rightarrow c_{\eta_u} + 2\theta, and c_{\eta_w} \rightarrow c_{\eta_w} + 2\theta$, which matches the abc symmetries identified in Yang et al.
>
> However, it is important to emphasize that Table 1 is not covered by the abc symmetries because the abc symmetries mainly address when changes in parameterization do not affect learning dynamics, while Table 1 mainly focuses on how selecting appropriate learning rates and normalizations can position the model in either the feature learning phase or the kernel phase. Moreover, Table 1 is not covered by the abc parameterization either,  because our results apply to **finite-width networks with arbitrary initialization**, whereas Tensor programs (abc parameterization) assume infinite width and Gaussian initialization, as discussed after Corollaries 1-3 in our paper.
>
> # Response to Comment 3
>
> * *The authors claim that they characterize the role of width. How does width enter their formulas, is it primarily through the initial alignment? Or is width to be interpreted as a possible scaling quantity?*
>
> Thank you for your question. In our analysis, width plays a role in two distinct ways. In Section 3, we focus on how width affects alignment, as seen in equation (14). Here, we show that initial alignment may remain zero for certain scalings when the width $d$ is sufficiently large (as discussed after equation (14)). Meanwhile, in Section 4, width is treated as a scaling quantity, where $d \propto \kappa^{c_d}​$, influencing whether the model operates in the feature learning or kernel phase (see equation (19)). Sections 4.1 and 4.2 explore specific cases with $c_d = 1$ and $c_d = $0, respectively.

---

> ### Author Response · Authors · 2024-11-16
>
> # Response to Comment 4
>
> * *Could Figure 4 be plotted on a log-log scale? I suspect that |W-W0|/|W0| should scale as a power law in d with exponent that is different for different parameterizations.*
>
> Thank you for the suggestion. We have replotted Figure 4 on a log-log scale, as recommended. Indeed, we observe that $|W - W_0| / |W_0|$ approximately follows a power-law scaling in d, with exponents that align **quantitatively** with our theoretical predictions according to Theorem 1. For further details, please refer to Appendix B.4 in the revised manuscript.
>
> # Response to Comment 5
>
> * *In Figure 5, it appears that the best performance is not at very small weights, but rather there is some optimal intermediate weight scale. How do the authors reconcile this?*
>
> Thank you for your question. As noted in our response to your first question, our results emphasize that overly large initializations can lead to disalignment, which negatively impacts performance. However, this does not imply that smaller initialization is always better. There is a trade-off: very small initializations may lead to optimization difficulties (e.g., slower convergence), while overly large initializations can cause disalignment. Thus, there exists an optimal intermediate scale where performance is maximized.
>
> While our study does not focus on the precise tuning of hyperparameters, previous work (e.g., Yang et al., 2022) suggests that careful tuning is still necessary to achieve optimal performance, even when using theoretically guided scalings.
>
> # Response to Comment 6
>
> * *Generally throughout the paper some measured metrics are implicitly related to performance/generalization, without this connection being made clear. For example "In the kernel phase, the model norm diverges and the model alignment is always zero, which could be a hint of strong overfitting."*
>
> Thank you for your suggestion. To clarify, when we refer to the model’s performance, we primarily mean its generalization error, as demonstrated in Figures 3 and 4. However, we do not claim a direct, formal connection between generalization error and alignment; our analysis is primarily intuitive and empirical. The statement you referenced proposes a potential mechanism: when alignment does not increase during training (as in the kernel phase), the model may fail to learn meaningful features, which can lead to overfitting and, consequently, worse generalization. A more rigorous investigation of this mechanism is beyond the scope of our current work but would be a valuable direction for future research.

---

> ### Comment · Reviewer_Wsc1 · 2024-11-21
>
> I thank the authors for their detailed rebuttal and updates to the paper. My one remaining suggestion would be to add a caveat sentence about the relationship between alignment and generalization being beyond the scope of the current study. I will increase my score.

---

> > ### Author Response · Authors · 2024-11-22
> >
> > Thank you for your reply! We will surely follow your suggestion in the final version.

---

### Author Response · Authors · 2024-11-16
**Rebuttal by Authors**

Dear Reviewers,

We thank all the reviewers for their constructive feedback. We are encouraged to hear that the reviewers recognize the importance of our exact solutions for finite-width models and appreciate the generality of our theory (e.g., “This paper studies an important problem of how model initialization impacts learning dynamics in neural networks” by **Wsc1**, and “The obtained parameterizations are quite general, covering various width, learning rate, and initialization scales” by **T1qw**).

It is particularly encouraging to hear that the reviewers found that our results provide novel insights into the mechanisms of feature learning, which was the primary objective of our study (e.g., “The notion that the feature learning component can come from three distinct 'types of feature learning' was new and interesting to me” by **jPRE**, and “The paper is quite thorough and does a great job covering the mechanisms of feature learning in this toy setting” by **9ybf**).

We have carefully incorporated the reviewers' suggestions, which have significantly enhanced the clarity and quality of our manuscript. Below, we outline how we addressed the main concerns raised:

* Clarification on the initialization scale (**Wsc1**,**jPRE**). See updates in Section 3.4 and our explanations in the rebuttal.
* Power-law scaling of weight evolution (**Wsc1**).See updates in Section 4.1 and Appendix B.4.
* Comparison with catapult (uv) model (**T1qw**). See updates in Appendix A.
* Clarification that Corollaries 1-3 also hold for finite-width networks (**T1qw**). See updates in Section 4.

We have highlighted these revisions in the manuscript for clarity (marked in red). We believe the updated version addresses all concerns and significantly strengthens our work. We welcome any further discussion or questions the reviewers may have.

Thank you once again for your valuable feedback.

Best regards,

The Authors

---

### Comment · Area_Chair_PyGA · 2024-11-19
**Discussion Phase**

Dear Reviewers,

Please review the authors' replies and consider the feedback from other reviewers. If your concerns remain unresolved, feel free to ask for further clarifications. We have until November 26th for discussion.

Thank you for your efforts.

Best regards,
Area Chair

---

> ### Comment · Area_Chair_PyGA · 2024-11-23
>
> Dear Reviewers,
>
> As we near the end of the discussion period, this is your last chance to engage.
>
> Please review the authors' replies and feedback from other reviewers. If any concerns remain, request clarifications. The authors have one more day to respond before the reviewer-only discussion week begins.
>
> Thank you for your efforts.
>
> Best regards,
> Area Chair

---

### Meta-Review · Area_Chair_PyGA · 2024-12-19

**Metareview:**

### Summary

The paper proposes a theoretical framework to investigate different regimes of learning.
The authors proposed a novel model that allows a solvable solution.
The authors used this framework to interpolate between feature and lazy learning regimes
and characterised three mechanisms of learning within their framework.

### Strengths

The paper is very clear. The authors made a great job in presenting the theory and explaining the insights of their equations.
Focusing on the technical contributions: there are very few theoretical frameworks that allow us to analyse the transitions from feature learning to lazy learning (spanning several of the limits considered in other works) and even fewer are amenable to an explicit solution.
On top of this, the results from the analyses are very interpretable and
the authors managed to identify 3 mechanisms emerging from the equations.
The theory is based on a simplified model of learning that, however, shows some degree of universality.
Indeed the authors provided numerical experiments confirming some of the predictions reported in their study.

### Weaknesses

The main weakness is the simplicity of the theory.
The authors effectively study a 1-dimensional input which is far from a realistic setting.
The theory should allow explicit solutions for polynomial activations with exponents \beta = 1, 2, and 3,
however the paper analyses only the linear case (\beta = 1) because the other exponents presented solutions hard to interpret.
In the words of the authors: "it is difficult to write the results in a comprehensible form".
Effectively this is a linear theory for 1-dimensional inputs.

### Reasons for acceptance

Despite the simplicity of the model, the theory is solid and shows some degrees of universality in more complex applications.
The Reviewers (and the AC) agree on the paper's relevance and consider it an important contribution that should be presented at the conference.

**Additional Comments On Reviewer Discussion:**

Most of the reviewers' comments focused on clarifications regarding various aspects of the framework. While the theory is interpretable, the notation didn’t always make key dependencies immediately clear. The authors provided satisfactory replies, and the comments were incorporated into the revised version.

Other feedback pertained to missing references, which were also integrated into the final submission.

By the end of the discussion, the reviewers were aligned in their recommendation for acceptance.

---

### Decision · Program_Chairs · 2025-01-22

Accept (Poster)